# ⏱ TiC-CLIP: Continual Training of CLIP Models

**Saurabh Garg**[‡*]   **Mehrdad Farajtabar**[†]   **Hadi Pouransari**[†]   **Raviteja Vemulapalli**[†]

**Sachin Mehta**[†]   **Oncel Tuzel**[†]   **Vaishaal Shankar**[†]   **Fartash Faghri**[†]

[†]Apple [‡]Carnegie Mellon University
sgarg2@andrew.cmu.edu, fartash@apple.com

## Abstract

Keeping large foundation models up to date on latest data is inherently expensive. To avoid the prohibitive costs of constantly retraining, it is imperative to *continually* train these models. This problem is exacerbated by the lack of any large scale continual learning benchmarks or baselines. We introduce the first set of web-scale Time-Continual (TiC) benchmarks for training vision-language models: TiC-DataComp, TiC-YFCC, and TiC-RedCaps. TiC-DataComp, our largest dataset, contains over 12.7B timestamped image-text pairs spanning 9 years (2014–2022). We first use our benchmarks to curate various *dynamic* evaluations to measure temporal robustness of existing models. We show OpenAI's CLIP (trained on data up to 2020) loses $\approx 8\%$ zero-shot accuracy on our curated retrieval task from 2021–2022 compared with more recently trained models in OpenCLIP repository. We then study how to efficiently train models on time-continuous data. We demonstrate that a simple rehearsal-based approach that continues training from the last checkpoint and replays old data reduces compute by $2.5\times$ when compared to the standard practice of retraining from scratch[1].

## 1 Introduction

Large multimodal foundation models (Bommasani et al., 2021) have offered unprecedented advancements in image-generation and zero-shot generalization, and have led to a paradigm shift in multimodal learning, e.g., CLIP (Radford et al., 2021), Flamingo (Alayrac et al., 2022), and Stable Diffusion (Rombach et al., 2022). These foundation models are typically trained on large web-scale datasets which are fixed and *static* in nature. For example, CLIP's training data contains 400 million image-text pairs, and Stable Diffusion was trained on LAION-2B dataset (Schuhmann et al., 2022). In reality, however, these models must operate in a *dynamic* environment, where the world is in a state of constant change. For instance, the internet continually evolves, with petabytes of new data being added daily (Wenzek et al., 2019; Wiener & Bronson, 2014). It remains unclear how legacy models, e.g., OpenAI's CLIP models which were trained on internet-scale data up until 2020, work on future data and whether they even require any re-training to adapt to time-evolving data.

We begin by comparing robustness of OpenAI's CLIP models to others in OpenCLIP repository that are trained on more recently curated web-datasets (e.g., LAION-5B, DataComp) containing data up until 2022 (Ilharco et al., 2021). Since there is no existing benchmark to understand robustness to time-evolving vision-language data, we curate *dynamic* classification and retrieval tasks for years 2014–2022 and evaluate different CLIP models (see Sec. 2.2 for our evaluation tasks). We make an intriguing observation that OpenAI models exhibit a significant gap in retrieval performance on data from 2021–2022 compared with 2014–2016 whereas OpenCLIP models retain their performance. In contrast, standard evaluations such as accuracy on ImageNet distribution shifts paint an incomplete picture that OpenAI's CLIP models are slightly more robust than OpenCLIP models (Fig. 1). Our findings not only demonstrate the critical need for models to adapt and evolve alongside dynamic data distributions, but also underscores the limitations of relying solely on static benchmarks (e.g. ImageNet).

One naive but common practice for adapting to time-evolving data is to train a new CLIP model from *scratch* every time we obtain a new pool of image-text data. This practice has its rationale:

---

*Work done during an internship at Apple.

[1]Code is available at https://github.com/apple/ml-tic-clip.

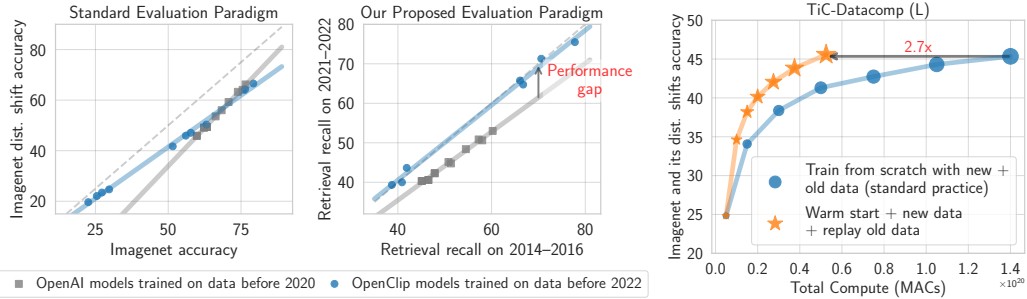

Figure 1: *(Left, Middle)* **OpenAI models show less zero-shot robustness on retrieval task from 2021–2022.** OpenCLIP models and OpenAI models have similar robustness on standard benchmarks. However, OpenAI models show less robustness on our retrieval task when compared with recent models in OpenCLIP repository, highlighting susceptibility to a time-evolving data distribution *(Right)* **Simple continual training baseline is computationally efficient and competitive to retraining from scratch.** Different points denote models trained sequentially on our TIC-DataComp (L) as data arrives over time. Warm start training with previous checkpoint and replaying all old data, performs similar to Oracle which trains from scratch every time new data arrives, by using 2.7× less compute.

initiating training from a pre-existing model can make it difficult to change the model's behavior in light of new data (Ash & Adams, 2020; Achille et al., 2018; Liu et al., 2023). However, training foundation models from scratch demands significant computational resources and is often infeasible to repeat frequently. For example, ViT-g-14 in Schuhmann et al. (2022); Cherti et al. (2022) was trained for 240K A100 GPU hours which is approximately one month on 400 GPUs. The prevailing training guidelines centered around scaling laws for CLIP training have only looked at training from scratch (Cherti et al., 2023). This leads to a pivotal question: *How can we continuously update models as the data distribution evolves over time given computational constraints?*

There exists a vast literature on continual learning, with a focus on adapting models to dynamic environments (Parisi et al., 2019; Hadsell et al., 2020; De Lange et al., 2021). Traditionally, this field concentrated on synthetic incremental benchmarks that lack natural evolution between tasks, and hence, continual learning methods are seldom used in real-world scenarios (Cossu et al., 2022; Lin et al., 2021). In contrast, recent works focusing on continual learning methods for CLIP models, primarily target improving performance on a single or a sequence of disjoint downstream tasks (Ding et al., 2022; Zhou et al., 2023b; Zheng et al., 2023; Ilharco et al., 2022). While some recent works have started to address these problems, existing benchmarks are comparatively much smaller in scale, or lack paired image-text data (Ni et al., 2023; Lin et al., 2021). Simply put, there is a scarcity of work focusing on continual training of CLIP models on naturally evolving data with time at web-scale.

We take the first step towards **Time-Continual (TIC)** training of CLIP models where data distribution evolves naturally over time (overview in Fig. 2). We introduce TIC-DataComp, a new benchmark for Time-Continual training of CLIP models, which we create by appending "crawl time" information to existing CommonPool dataset (Gadre et al., 2023). We also repurpose other web-scale datasets gathered from diverse sources, such as Reddit and Flickr. Specifically, we curate TIC-YFCC and TIC-RedCaps by leveraging time information available in YFCC (Thomee et al., 2016) and Redcaps (Desai et al., 2021) respectively. The primary objective of our study on this benchmark is to develop continual learning methods that operate within a constrained computational budget (say $C$) each time a fresh batch of data becomes available. These methods compete with an Oracle, which starts training from scratch every time new data arrives, utilizing a cumulative computational budget.

To assess models trained in our TIC-CLIP framework, we evaluate models on our proposed dynamic evaluation tasks that evolve with time along with 28 standard classification and retrieval tasks including ImageNet (Krizhevsky et al., 2012), ImageNet distributions shifts, and Flickr (Plummer et al., 2015), in a zero-shot manner following the work of Gadre et al. (2023); Radford et al. (2021).

Finally, we develop continual learning methods on our benchmarks and perform over two hundred experiments with different baselines that utilize previous checkpoints (e.g., warm start, patching, and distillation), replay buffers, and learning rate schedules. Our findings highlight a key takeaway: Cumulative method that warm starts training with the latest checkpoint and replays all old data, achieves performance competitive to an Oracle while being 2.7× computationally more efficient.

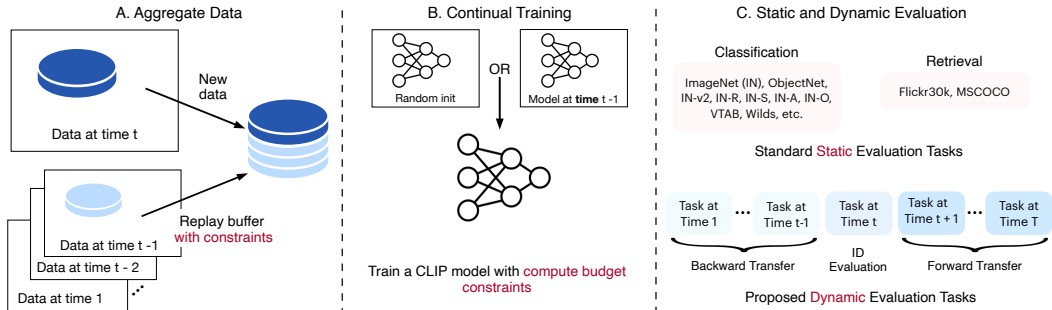

Figure 2: **Experimental protocol on our proposed continual benchmarks.** *(A)* Combine new and old data given buffer constraints. *(B)* Continually train a model with a compute budget (say $C$) either by starting with previous checkpoint or from scratch. *(C)* Evaluate models on standard datasets and our proposed dynamic datasets. Comparison with other benchmarks in Appendix A.

Additionally, our experiments demonstrate interesting trade-offs between buffer sizes for static and dynamic performance and provide valuable insights into learning rate schedules for sequential training. Our results span over various dataset scales (from 11M samples to 3B) and highlight trends with different methods that are largely consistent across scales.

To make our benchmarks accessible, we publicly release the code and the time information we collect on top of existing datasets here. Our work is just an initial step towards continual training of foundation models, and we believe our research would spur more attention to this understudied area.

## 2 TIC-CLIP: BENCHMARKS AND EXPERIMENTAL PROTOCOL

In this section, we introduce our benchmark (Fig. 2) focusing on the training of a vision-language foundation model with the Contrastive Language Image Pretraining (CLIP) (Radford et al., 2021)) objective. Notably, we train on image-text data that arrives sequentially unlike the conventional image-text datasets which are static (e.g. WiT in CLIP, DataComp in Gadre et al. (2023)). We curate TIC-DataComp, TIC-YFCC, and TIC-RedCaps that are image-text pairs sourced from the internet which we augment with auxiliary time information. We also introduce dynamic evaluation tasks to assess performance of our continually trained models on data evolving with time. The goal of a learner is to train a *deployable* model at each step as new data becomes available with a fixed compute budget.

### 2.1 BENCHMARK DESIGN: HOW WE CREATE TIME-CONTINUAL DATASETS?

To instantiate continual training of CLIP, we extend existing image-text datasets with time information collected from the original source of the datasets. Our largest dataset is TIC-DataComp which contains 12.7 billion image-text pairs with "crawl-time" metadata. We create this dataset on top of the existing DataComp benchmark (Gadre et al., 2023). We also create TIC-YFCC and TIC-RedCaps on top of existing YFCC15M (Thomee et al., 2016; Radford et al., 2021) and Redcaps (Desai et al., 2021) datasets to highlight that our findings are broadly applicable to carefully curated datasets from diverse sources such as Reddit and Flickr. While time-related metadata is absent in the DataComp benchmark, it is available in the original releases of YFCC and Redcaps. Nevertheless, to the best of our knowledge, no prior work utilizes such time information for continual training of CLIP models. We show dataset statistics for all datasets, e.g., number of examples in each year in App. C.3.

**TIC-DataComp** We collect timestamps for the CommonPool dataset introduced in DataComp which contains 12.7B image-text pairs (not including 0.1B inaccessible ones). This dataset stands as the largest public image-text dataset to date. The source of DataComp is Common Crawl, which periodically releases web-crawled data snapshots, typically on a monthly basis since 2014 with new and updated webpages. To construct TIC-DataComp, we augment each image-text pair in DataComp with their *first* timestamp. We followed the same construction process as DataComp but retained only the image-text pair found in the earliest snapshot during the deduplication stage. This process provides timestamps at the granularity of months, spanning years 2014–2022. See App. C.7 for details on the construction process. We note that while this augmented time information may contain some noise, on average, we find it to be a reasonably accurate proxy for the upload time of web pages (see App. C.7).

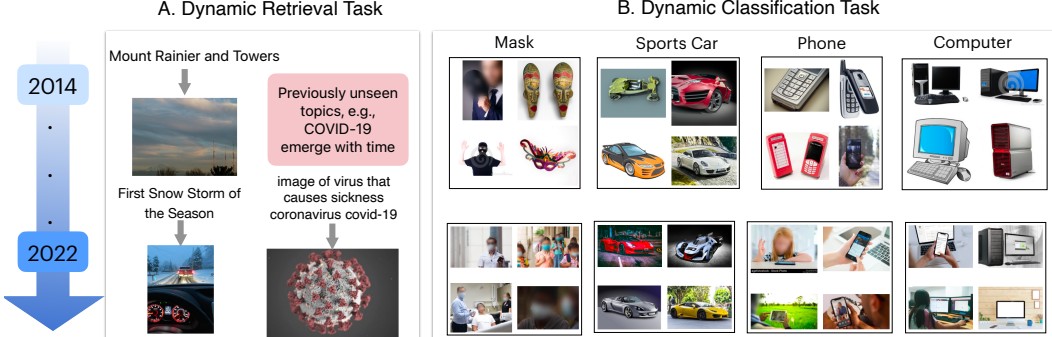

Figure 3: **Distribution of examples changes from 2014 to 2022 in our dynamic evaluation tasks.** *(Left)* Samples for text to image retrieval. For new timestamps, images from novel concepts appear (e.g., COVID-19). *(Right)* Samples from our classification task for 4 categories. We observe that not only objects evolve over time but also images from recent timestamps are captured more in the wild.

Although our benchmark contains time information at the granularity of months, we limit our experiments to granularity of years by consolidating data for all months in a year. Similar to DataComp, our benchmark has an inclusive design, accommodating participants with varying levels of computational resources. In particular, we experiment with `medium`, `large`, and `xlarge` sizes from CommonPool. Gadre et al. (2023) leverage different filtering strategies to select the training subset. We are concerned that filtering techniques bias the selected training data. In App. C.1, we provide preliminary evidence that "Bestpool" filtering that uses off-the-shelf CLIP models, indeed biases the selected data to old time steps. Nevertheless, to highlight significance of our findings even for state-of-the filtering techniques, we experiment with both Bestpool and Basic filtering (no CLIP filtering) at `xlarge` scale. For `large` and `medium` scales, we only experiment with Basic filtering.

**TıC-YFCC** We experiment with the 15M subset of YFCC100M (Thomee et al., 2016), namely YFCC15M, selected by OpenAI (Radford et al., 2021). This filtering retains only images with natural text in captions. YFCC100M contains data from years 2008–2014 and was originally released with upload timestamps. We use this information to create continual splits at the granularity of years.

**TıC-RedCaps** RedCaps contains 12M image-caption pairs from manually curated set of subreddits across 2011–2020 (Desai et al., 2021). We use the creation timestamps of the posts to create splits for continual learning. Similar to the other two datasets, we experiment at the granularity of years.

## 2.2 Evaluation Testbed

**Dynamic tasks** We leverage the temporal information in our benchmarks to create *dynamic evaluation* tasks. Here, the test data comprises samples varying over years as the world evolved. For our largest dataset which is TıC-DataComp, we create dynamic tasks for both retrieval and classification as described below. (examples in Figure 3 and additional examples in App. C.5):

I. *Dynamic retrieval task*: To create a retrieval task, we sample a batch of IID image-text pairs from different timestamps and evaluate text retrieval performance given the corresponding image (similarly, image retrieval given the corresponding text). We refer to the dataset as TıC-DataComp-Retrieval.

II. *Dynamic classification task*: We also create a classification dataset TıC-DataComp-Net with ImageNet classes from CommonPool and augmented with timestamps. Inspired by LAIONNet (Shirali & Hardt, 2023), we first filter examples where the corresponding caption contains one and only one of the synsets of ImageNet. Then we only retain examples where the similarity between ImageNet synset definition and the caption exceeds a threshold of 0.5. We evaluate the similarity using an off-the-shelf sentence embedding model (Reimers & Gurevych, 2019). Crucially, unlike LAIONNet, we do not filter the image-text pairs with CLIP similarity scores to avoid biasing the selection process. We describe the construction process in more details in App. C.5. On TıC-DataComp-Net, we report average accuracy over all classes and over selected nodes (e.g., motor vehicles) at each time step.

Similarly, we create retrieval tasks for TıC-YFCC and TıC-RedCaps. Note that we remove the extracted image-text pairs for dynamic retrieval and classification tasks from the training sets. Evaluations on dynamic tasks are done in a zero shot manner.

**Static tasks**   We also evaluate models on numerous classification and retrieval tasks in a zero-shot manner as in Radford et al. (2021). In particular, we consider 28 standard tasks: 27 image classification tasks, e.g., ImageNet and its 6 distribution shifts (e.g., ImageNetv2, ImageNet-R, ImageNet-Sketch, and Objectnet), datasets from VTAB and Flickr30k retrieval task. We refer to these as *static evaluation* tasks. We list all the datasets in App. C.2.

**Evaluation metrics**   We define metrics for classification tasks and retrieval tasks based on *accuracy* and *Recall@1*, respectively. Let $T$ represent the number of time steps for which we have data. For each training method, we generate a total of $T$ models, each corresponding to the end of training at a particular time step. For static datasets (e.g., ImageNet), we report average performance of $T$ models. However, when dealing with dynamic evaluation datasets, we assess the performance of each of the $T$ models on evaluation datasets collected at all time steps. Consequently, for each model and a dynamic evaluation task, we obtain $T$ performance values. We represent these values using the performance matrix $\mathcal{E}$, where each entry $\mathcal{E}_{i,j}$ signifies the performance of the model obtained after observing training data at time step $i$ when evaluated on a dataset from time step $j$. The performance matrix $\mathcal{E}$ can also be succinctly summarized using three standard metrics commonly employed in continual learning evaluations (Lin et al., 2021; Díaz-Rodríguez et al., 2018):

• *In-domain performance*: average performance at each training time step (i.e., the diagonal of $\mathcal{E}$)

• *Backward transfer*: average on time steps before each training step (i.e., the lower triangular of $\mathcal{E}$)

• *Forward transfer*: average on time steps following each training step (i.e., the upper triangular of $\mathcal{E}$)

Sometimes, the metrics described above can cause the backward transfer metric to be influenced by later evaluation time steps, biasing the backward transfer metric (refer to App. F for details). Therefore, in App. F, we present results using revised metrics that mitigate this issue.

While the static tasks capture performance on standard benchmarks, dynamic tasks capture problems due to distribution shift (for forward transfer) and forgetting (for backward transfer). The goal in our benchmark is to develop continual learning methods that maximize performance on static tasks while simultaneously optimizing for performance on dynamic tasks.

## 2.3   EXPERIMENTAL PROTOCOL FOR TRAINING

**Streaming protocol**   We follow a streaming protocol, where data is progressively revealed to the learner in large batches with the objective of achieving a deployable model as early as possible after each batch arrives. We conduct experiments with data streaming at the granularity of years and our benchmark supports future research at the granularity of months. Additionally, as the amount of data from earlier time steps is limited (see App. C.3), we aggregate data from the earlier time steps into a single larger batch and timestamp it by the latest year in the range. After this aggregation, we have 7 time steps for TIC-DataComp (2016–2022) and 4 for both TIC-YFCC (2011–2014) and TIC-RedCaps (2017–2020). While the number of image-text pairs revealed at each time step are of similar orders of magnitude, the exact number does vary across steps and we do not artificially alter the sizes.

**Memory budget**   We allow methods to use the last model checkpoint at each step as the cost of keeping one checkpoint per month is often negligible. In contrast, the cost of retaining old data can be high and might not be permitted due to data expiration policies. Thus, along with studying methods that retain all old data, we also explore strategies that restrict data persistence (see Sec. 3 for details).

**Compute budget**   To ensure a fair comparison between methods, we establish a consistent total compute budget, quantified in terms of Multiply-Accumulate Operations (MACs), and allocate it evenly for training at every time step. Unless specified otherwise, for all methods except Oracle and LwF, we use the same compute budget. For experiments on TIC-DataComp, we refer to compute configurations in DATACOMP for overall compute. For TIC-RedCaps and TIC-YFCC, we use compute of order `medium` scale in TIC-DataComp. Compute budget details are in App. C.4.

## 2.4   ANALYZING DISTRIBUTION SHIFTS IN THE CONSTRUCTED BENCHMARKS

**TIC-DataComp analysis through the lens of constructed evaluation tasks**   First, we qualitatively analyze the examples in our retrieval and classification dataset (Fig. 3). We observe that over time, in the retrieval task, new concepts like COVID-19 emerge. Likewise, certain ImageNet classes evolve, such as the shift from "masquerad" masks to "surgical/protective" masks in their definitions. Moreover, as time evolves, we observe that image quality improves and more images tend to appear in the wild in contrast to centered white background images. Next, we compare performance of OpenAI

and OpenCLIP models on our datasets. Here, we only present the main findings, and delegate a detailed discussion to App. C.6. We observe a significant performance gap between OpenAI and OpenCLIP models on our dynamic retrieval task (Fig. 1). This gap widens notably on retrieval queries where captions mention COVID-19. On the other hand, OpenAI and OpenCLIP models exhibit similar robustness for retrieval on data coming from Flickr highlighting that data from some domains do not exhibit shifts that cause performance drops. For our classification task, we observe a very small drop ($\approx 1\%$) when averaged across all categories. However, we observe a substantial gap on specific subtrees in ImageNet. For example, classes in "motor vehicle" subtree show an approximate $4\%$ performance drop, when comparing OpenAI and OpenCLIP models. These findings highlight that while overall ImageNet classes may remain timeless, certain categories tend to evolve faster than others. Our qualitative and quantitative analysis on TIC-DataComp clearly highlights evolution of distributions and captures different properties than standard benchmarks.

**Quantitative analysis on TIC-YFCC**  We analyze TIC-YFCC using off-the-shelf sentence and image encoders. We first embed images from different time steps with an OpenAI CLIP encoder and then compute Frechet Inception Distance (FID; Seitzer (2020)). As time progresses, we observe that FID distance increases with respect to data from first time step (Fig. 18 in App. C.6). Similarly, we use pretrained sentence transformer to extract top-5 categories from Wordnet Nouns for each caption. We observe that the TV distance over distribution of WordNet Nouns evolves over time when compared to data from the first time step. More details in App. C.6.

## 3 TIC-CLIP: How to Continually Train CLIP Models?

In this section, we lay out different methods specifically focus on the following questions (Tab. 1): (i) How to utilize/replay data from previous time steps; (ii) How to leverage previously trained model checkpoints? (iii) What should be the training/optimization procedure?

Data replay methods initialized from the last checkpoint demonstrate strong performance on standard continual learning benchmarks (Sec. 5). We consider replay methods with/without initialization from last checkpoint(s):

Table 1: Table summarizing our methods. $D$: data size in each step, $T$ total time steps, $t$: current time step, $C$: compute budget (iterations).

| Method | Each Step | | | Total |
|---|---|---|---|---|
| | Train Size | Init. | Compute | Compute |
| Cumulative-All | $tD$ | Last | $C$ | $TC$ |
| Cumulative-Exp | $2D$ | Last | $C$ | $TC$ |
| Cumulative-Equal | $2D$ | Last | $C$ | $TC$ |
| Sequential | $D$ | Last | $C$ | $TC$ |
| Restart | $tD$ | Rand | $C$ | $TC$ |
| Patching | $D$ | Last Patch | $C$ | $TC$ |
| LwF | $D$ | Last | $1.2 \times C$ | $1.2 \times TC$ |
| Oracle** | $tD$ | Rand | $tC$ | $\frac{(T+1)T}{2}C$ |

I. **Oracle**: Train a CLIP model from scratch (i.e., random initialization) on all image-text data received till time $t$ using a large compute budget of $t \times C$. Oracle represents a *prohibitively expensive* method that is the most common practice in training large-scale foundation models. The goal of other methods is to perform as close as possible to the Oracle within their limited budget.

II. **Cumulative**: Train each model initialized from last checkpoint on the union of all data up to $t$ with compute budget $C$. This method is analogous to experience replay (Robins, 1995; Hayes et al., 2019) but with substantially larger buffers than common in the continual learning literature. Given a fixed buffer size for each past step, we observe minimal to no difference between random subsampling and other strategies. After sampling the replay data, we randomly shuffle it together with new data for training. We consider the following strategies for sampling buffer sizes per step:

- **-All**: Replay all previous data.
- **-Exp**: Replay a buffer of size $D$ and reduce the amount of old data by half at each step. For example, at 3-rd time step, we retain $D/2, D/2$ of old data and at 4-th, we retain $D/4, D/4, D/2$ of old data. Along with $D$ data from current step, this method trains on at most $2D$ data in each step.
- **-Equal**: Replay a buffer of size $D$ but split the buffer equally among all previous years. For example, at 4-th step, we retain $D/3, D/3, D/3$ of old data. Along with $D$ data from current time step, this method trains on at most $2D$ data in each step.

III. **Sequential**: Train *only* on the new data starting from the best checkpoint of the previous time step. Sequential is similar to Cumulative but without any replay buffer.

IV. **Restart**: Train each model from scratch (i.e., random initialization) on all the data till time $t$ for compute budget $C$. Restart is similar to the Oracle but with compute budget $C$ at each time step and similar to Sequential but with random initialization. As such, Restart helps us understand the *forward transfer* and *loss of plasticity* in our benchmark (Ash & Adams, 2020; Dohare et al., 2023).

Table 2: **Zero shot performance on our time-continual benchmarks.** * and ** denote methods that violate the compute budget. For static tasks, we tabulate accuracy of the models obtained on the final timestamp. For dynamic tasks, we tabulate forward/backward transfer and ID performance on retrieval tasks (Sec. 2.3). For TiC-DataComp (XL), we include results with Bestpool filtering (basic filtering in Table 5). For all metrics, higher is better.

| Benchmark | Method | Compute (MACs) | Static Tasks | | | | Dynamic Retrieval Tasks | | |
|---|---|---|---|---|---|---|---|---|---|
| | | | ImageNet | ImageNet dist. shift | Flickr30k | Average over 28 datasets | Backward Transfer | ID Performance | Forward Transfer |
| **TiC-YFCC** | Restart | $3.4 \times 10^{18}$ | 5.2 | 3.6 | 3.0 | 12.9 | 13.2 | 41.4 | 18.6 |
| | Sequential | $3.4 \times 10^{18}$ | 17.3 | 10.5 | 15.9 | 21.9 | 42.2 | 48.4 | 23.7 |
| | Patching | $3.4 \times 10^{18}$ | 18.9 | 11.3 | 18.5 | 23.3 | 44.7 | 53.4 | 24.5 |
| | Cumulative-Exp | $3.4 \times 10^{18}$ | 24.1 | 14.3 | 20.4 | 25.9 | 60.4 | 60.1 | 27.1 |
| | Cumulative-Equal | $3.4 \times 10^{18}$ | 23.9 | 13.8 | 20.5 | 26.3 | 60.4 | 60.4 | 27.1 |
| | Cumulative-All | $3.4 \times 10^{18}$ | **29.3** | **17.6** | **26.8** | **29.6** | 66.4 | 60.2 | **27.6** |
| | LwF* | $4.1 \times 10^{18}$ | 16.9 | 9.8 | 14.7 | 21.2 | 36.6 | 56.0 | 23.2 |
| | Cumulative-All* | $3.6 \times 10^{18}$ | **29.2** | **17.5** | **27.4** | **29.3** | **66.8** | **60.3** | **27.6** |
| | Oracle** | $8.5 \times 10^{18}$ | **29.2** | **17.0** | **25.9** | **29.0** | 66.1 | 61.8 | 26.9 |
| **TiC-RedCaps** | Restart | $3.4 \times 10^{18}$ | 11.7 | 8.5 | 3.7 | 18.4 | 21.3 | 25.4 | 22.4 |
| | Sequential | $3.4 \times 10^{18}$ | 19.3 | 13.7 | 6.2 | 25.8 | 33.0 | 33.6 | 27.5 |
| | Patching | $3.4 \times 10^{18}$ | 21.3 | 15.2 | 7.7 | 26.8 | 34.8 | 34.8 | 27.8 |
| | Cumulative-Exp | $3.4 \times 10^{18}$ | 27.3 | 19.1 | 10.5 | 30.0 | 44.5 | 42.0 | 32.6 |
| | Cumulative-Equal | $3.4 \times 10^{18}$ | 27.8 | 19.4 | 10.0 | 30.5 | 44.4 | 42.0 | 32.6 |
| | Cumulative-All | $3.4 \times 10^{18}$ | **32.2** | 18.7 | **14.5** | 31.7 | **48.9** | **43.2** | **33.4** |
| | LwF* | $4.1 \times 10^{18}$ | 21.6 | 14.8 | 8.2 | 27.3 | 35.4 | 36.0 | 28.4 |
| | Cumulative-All* | $3.6 \times 10^{18}$ | **32.9** | **23.7** | **14.1** | **32.9** | **49.0** | **43.4** | **33.4** |
| | Oracle** | $8.5 \times 10^{18}$ | **32.7** | **22.7** | **14.3** | **32.3** | 48.5 | 43.1 | **33.4** |
| **TiC-DataComp** (M) | Sequential | $3.0 \times 10^{18}$ | 19.2 | 16.4 | 16.4 | 26.0 | 25.7 | 26.4 | 14.9 |
| | Patching | $3.0 \times 10^{18}$ | 19.3 | 16.8 | 18.5 | 26.4 | 26.9 | 25.4 | 14.5 |
| | Cumulative-Exp | $3.0 \times 10^{18}$ | 22.1 | 18.4 | 20.4 | 28.8 | 31.7 | 27.1 | **15.2** |
| | Cumulative-Equal | $3.0 \times 10^{18}$ | 22.1 | 18.4 | 19.2 | 28.0 | 31.8 | 26.8 | 15.1 |
| | Cumulative-All | $3.0 \times 10^{18}$ | 24.0 | 20.2 | 20.9 | 30.0 | 33.8 | 26.4 | 15.1 |
| | LwF* | $3.8 \times 10^{18}$ | 19.2 | 16.5 | 17.7 | 27.0 | 25.6 | 26.6 | 14.9 |
| | Cumulative-All* | $3.9 \times 10^{18}$ | **30.0** | **25.0** | **28.6** | **35.1** | **36.7** | **28.3** | **15.5** |
| | Oracle** | $1.2 \times 10^{19}$ | 25.5 | 21.2 | 23.3 | 30.8 | 34.9 | 27.8 | **15.6** |
| **TiC-DataComp** (L) | Sequential | $2.7 \times 10^{19}$ | 44.7 | 37.4 | 48.4 | 45.7 | 52.6 | **58.4** | 41.1 |
| | Patching | $2.7 \times 10^{19}$ | 45.8 | 38.9 | 49.7 | 46.9 | 55.2 | 57.5 | 40.9 |
| | Cumulative-Exp | $2.7 \times 10^{19}$ | 47.3 | 39.6 | 50.8 | 47.6 | 60.4 | **58.4** | **41.4** |
| | Cumulative-Equal | $2.7 \times 10^{19}$ | 47.7 | 40.3 | 51.8 | 47.7 | 60.9 | 58.2 | **41.4** |
| | Cumulative-All | $2.7 \times 10^{19}$ | 48.9 | 41.3 | 50.9 | 48.0 | 62.1 | 57.3 | 41.2 |
| | Cumulative-All* | $4.1 \times 10^{19}$ | 53.0 | **44.3** | **54.4** | **51.3** | 63.0 | 57.8 | 41.2 |
| | Oracle** | $1.1 \times 10^{20}$ | **53.6** | 44.0 | 53.9 | 50.4 | **64.3** | **58.6** | **41.8** |
| **TiC-DataComp** (XL) | Sequential | $2.7 \times 10^{20}$ | 66.5 | 54.2 | 61.2 | 61.0 | 63.1 | 68.9 | 56.8 |
| | Cumulative-All | $2.7 \times 10^{20}$ | 71.6 | 58.8 | 65.1 | 64.8 | **70.7** | 68.5 | **57.1** |
| | Cumulative-All* | $3.5 \times 10^{20}$ | **72.8** | 60.4 | 66.5 | **66.7** | 71.0 | 68.6 | **57.1** |
| | Oracle** | $1.1 \times 10^{21}$ | **73.3** | **61.3** | **68.0** | 65.8 | - | - | - |

V. **Patching**: We use sequential patching from Ilharco et al. (2022). Initialize from a patched model of last step and train only on the new data. To obtain a patched model at each time step, we apply weight interpolation with the patched model (if any) trained at time step $t - 1$ and the model trained at time step $t$. We tune the mixing coefficients by optimizing average retrieval performance on previous tasks.

VI. **LwF**: Train only on the new data with a KL divergence penalty between the image-text similarity matrix of last checkpoint and current model on each batch (Li & Hoiem, 2017; Ding et al., 2022). See App. E for results with other continual learning methods, e.g., EWC (Kirkpatrick et al., 2017).

**Learning rate schedule** The defacto Learning Rate (LR) schedule for training CLIP models is an initial linear increase to a maximum value, i.e., warm up, followed by a cosine decay (Radford et al., 2021; Gadre et al., 2023). We default to using a cosine LR schedule for each sequential run, resulting in a cyclic schedule and observe a significant increase in training loss early in subsequent runs when the LR is high. However, as training progresses, we observe that the increased loss decreases at a faster rate (when compared to training from scratch) allowing us to train with cyclic schedules. We discuss this more and explore an alternate learning rate schedule in App. B.5.

**Other Training details and hyperparameters** Unless specified otherwise, we closely follow the original CLIP training recipe (Radford et al., 2021). We train the CLIP variant with ViT-B/16 as the image encoder (Dosovitskiy et al., 2020). All training and hyperparameters can be found in App. D.2.

## 4 EXPERIMENTS AND MAIN RESULTS

Our main results are in Table 2 and more detailed plots on each dataset are in App. B.1. Recall, our goal is compete with an Oracle that re-trains from scratch every time new data is observed, both on dynamic and static tasks, while being computationally efficient. Here, we summarize our key findings:

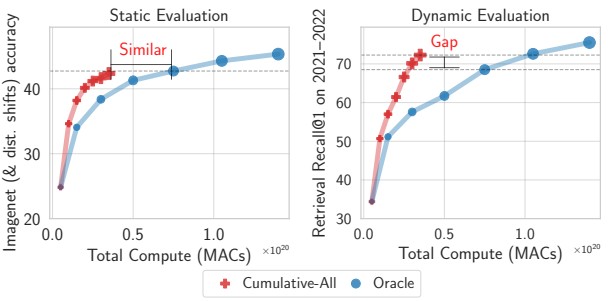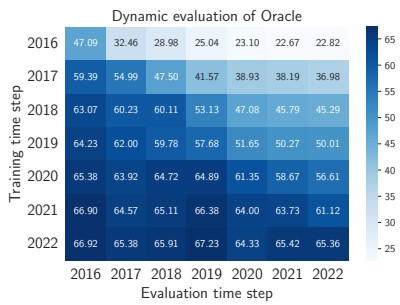

Figure 4: *(Left)* **Dynamic and static evaluations rank models differently**. Models with similar performance on static datasets, have $> 6\%$ difference on retrieval task from 2021-2022 TIC-DataComp (L). Different points denote models trained sequentially over time. *(Right)* **Performance of Oracle on future time steps drops highlighting distribution shift in dataset**. Each row evaluates the Oracle trained on TIC-DataComp (L) at a particular time step across all dynamic retrieval tasks.

**Cumulative-All saves up to $4\times$ the cost.** On dynamic evaluation tasks, we observe that Cumulative-All where we replay all the past data, achieves performance close to the Oracle (within 1%) using significantly less compute ($4\times$ less on TIC-DataComp and $2.5\times$ less on TIC-YFCC and TIC-RedCaps). On static tasks, the gap remains small at small scales but grows to 4.7% on `large`, 1.8% on `xlarge` Bestpool, and 4% on `xlarge` Basic (see Table 2 and Table 5). In these cases, training Cumulative models with slightly extra compute bridges the gap while remaining at least $2.7\times$ more computationally efficient (see rows with * in Table 2). This highlights that with unconstrained access to past data, we can simply train sequentially and save significant computational resources.

**At scale, Sequential has strong forward transfer but lacks on static tasks.** On TIC-YFCC and TIC-RedCaps, which are at the smallest scale, we observe a significant gap ($> 10\%$) between Sequential (with no data replay) and Oracle on all tasks. On the other hand, on all scales in TIC-DataComp, Sequential shows strong performance on forward transfer and ID dynamic evaluations. However, on static tasks and backward transfer evaluations, Sequential significantly underperforms the Oracle.

**Patching and LwF improve over Sequential but lag behind Cumulative-All.** On static tasks, LwF improves over Sequential by 2%, while on dynamic tasks, LwF improves backward transfer by 7% on TIC-DataComp (M). However, its computation cost is higher than even Cumulative-All* which outperforms LwF on all tasks. Patching improves over Sequential on backward transfer on all datasets (e.g., 5% boost on TIC-DataComp L) highlighting that Patching combines benefits of previously patched model and the new Sequential model without additional computation cost. However, such benefits do not show up on static tasks. These results hint that to continuously improve on static tasks with time, replaying old data as in Cumulative-All plays a crucial role.

**-Exp and -Equal significantly reduce replay buffer size and maintain static task performance and backward transfer.** Recall, that -Exp and -Equal reduce the replay buffer size to a maximum $2D$ of old data. In particular, at the last time step, -Exp and -Equal reduce the buffer size by $3.5\times$ for TIC-DataComp datasets. While reducing the buffer sizes, these methods still achieve performance close to Cumulative-All (within 2%) on both static and dynamic tasks, with -Equal consistently better than -Exp strategy. As we go to large scale, e.g., from `medium` to `large`, the gap between these methods and Cumulative-All reduces. These findings demonstrate that even a small amount of replay data from old time steps stays competitive with replaying all data and significantly improves over no replay at all.

**Warm up helps training on data from first time step, but hurts on subsequent time steps.** Cosine LR is commonly coupled with an initial warm-up that linearly increases the LR from zero to maximum LR. We investigate the effectiveness of warm-up in first versus subsequent time steps. Surprisingly, we observe that not using warmup for subsequent training runs is *strictly* more beneficial than using warm up on both static and dynamic tasks. In particular, on TIC-DataComp (L), we observe about 1.5% improvement in ImageNet accuracy and 4.3% improvement on ID dynamic retrieval when not using warmup with Cumulative (see App. B.3). Moreover, we also ablate over not using warm up for the first training run and observe a drop of approximately 4.8% accuracy in the first time step on TIC-DataComp (L). Hence, we default to using warmup when training on the first time step and not using it on the subsequent time steps with all methods except for training on TIC-DataComp (XL) where we add a smaller warm up (10% of the warm up iterations used in first step) to stabilize training.

**Same maximum LR works best across all runs when using cosine schedule.** We ablate on TIC-DataComp (M) to investigate how to change LR after training on data from the first time step. Unlike conventional pretraining and finetuning settings where LR is typically decreased for subsequent training, we observe that decaying maximum LR for subsequent steps in our setup hurts on static and dynamic tasks and consequently, we use same maximum LR across our runs (see App. B.3).

**Filtering strategy changes the ordering of performance on static and dynamic retrieval tasks.** We observe that while bestpool filtering models outperform basic filterining models on TIC-DataComp (XL) by 6% on static tasks, they underperform by over $5\%$ on dynamic retrieval task (see Fig. 7).

**Dynamic tasks provide complimentary information for model selection compared to static tasks.** Choosing models solely based on static task performance may inadvertently select models that underperform on dynamic tasks. For example, Cumulative models that show relatively modest improvements on static tasks continue to improve by $> 6\%$ for retrieval on 2021-2022 (Fig. 4).

**Cumulative-All remains competitive to Oracle even on ImageNet on up to 8 splits.** CLIP models are often trained for fewer epochs and are typically not trained until they reach an "overfitting" regime. Here, we investigate how Cumulative-All performs when compared to Oracle when training is done for longer. Specifically, we assess Cumulative-All on 2, 4 and 8 IID splits including the full dataset (see App. D.1 for details). Table 3 summmarizes our key findings. Notably, even with up to 8 splits, the difference in accuracy between Oracle and Cumulative-All remains below 0.9%. These results underscore the feasibility of continual training with Cumulative-All even on ImageNet.

Table 3: ImageNet continual training. Cumulative-All remains close to Oracle.

| Method | Number of splits | | | |
|---|---|---|---|---|
| | 1 (Oracle) | 2 | 4 | 8 |
| Cumulative-All | 80.9 | 80.8 | 80.6 | 80.0 |

## 5 RELATED WORK

**Benchmarks for continual learning** Traditionally, the continual learning community has focused on domain, class, and task incremental benchmarks (Hsu et al., 2018; Van de Ven & Tolias, 2019; Zhou et al., 2023a) with artificial task boundaries (e.g., Split-CIFAR, Perm-MNIST). These benchmarks are often task-specific and present minimal or no meaningful evolution between adjacent tasks. Consequently, continual learning methods are often confined to these benchmarks and seldom scale to practical real-world scenarios (Cossu et al., 2022; Lin et al., 2021). On the other hand, continual learning methods for CLIP models are primarily aimed at fine-tuning to improve performance on a single or on a sequence of disjoint downstream tasks (Thengane et al., 2022; Zheng et al., 2023; Ilharco et al., 2022). Existing large-scale benchmarks for training CLIP models, e.g., Datacomp (Gadre et al., 2023) and LAION-5B (Schuhmann et al., 2022), are curated to investigate methods and scaling laws to train state-of-the-art CLIP models in a single training run. In our work, we augment these existing datasets with temporal information to create benchmarks for continual pertaining of CLIP models.

**Continual learning methods** Common methods can be categorized into three categories: i) regularization, ii) replay, and iii) architecture-based methods. Regularization methods add a penalty to keep the fine-tuned model close to its initialization and often incur additional memory/compute costs (Kirkpatrick et al., 2017; Mirzadeh et al., 2020a;b; Farajtabar et al., 2020). Data replay methods retain all or a subset of the prior data for subsequent training (Lopez-Paz & Ranzato, 2017; Rebuffi et al., 2017; Chaudhry et al., 2018). Simple replay-based baselines surpass various methods on standard benchmarks (Lomonaco et al., 2022; Balaji et al., 2020; Prabhu et al., 2020). Lastly, architecture-based methods expand the model as new tasks arrive, limiting their applicability in evolving environments without clear task boundaries (Schwarz et al., 2018; Rusu et al., 2016). In this work, we compare popular continual learning methods with simple alternatives for continually pretraining of CLIP.

## 6 CONCLUSION AND FUTURE WORK

We view TIC-DataComp as the initial stride toward the continual training of large-scale vision-language foundation models. We believe that our benchmark, alongside the preliminary results obtained using simple baselines will foster future research for large-scale continual-learning. There are several pivotal directions for future work: (i) Compare our baselines on continually streaming data at finer granularity, e.g., streaming data at the monthly level; (ii) Investigate alternate learning rate schedules (e.g., Const-Cosine as in App. B.5) that are forward looking, and are better suited to continual learning; (iii) Better data filtering techniques that are more inclusive of future data; (iv) Expand our problem setup to encompass the training of other large-scale foundation models.

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

# A  CONTINUAL LEARNING BENCHMARKS AND METHODS

We introduce a large-scale image-text benchmark with web scale streaming image text pairs specially developed for studying how efficiently one can get a fresh CLIP model with new incoming batches of data. Table 4 compares the proposed benchmark with existing datasets for continual learning. Note that this table is not aimed to be an exhaustive list of all CL datasets, but the most popular benchmarks in each domain. For language modeling tasks we report the number of examples/documents as the number of samples and for detection tasks we report the number of labeled objects/bounding boxes.

Table 4: Comparison with continual learning benchmarks.

| Benchmark | # Samples | Years | Time-Continual | Image-Text | Task |
|---|---|---|---|---|---|
| Split-MNIST (Goodfellow et al., 2013) | 60K | 1998 | ✗ | ✗ | Classification |
| Perm-MNIST (Goodfellow et al., 2013) | 60K | 1998 | ✗ | ✗ | Classification |
| Rot-MNIST (Lopez-Paz & Ranzato, 2017) | 60K | 1998 | ✗ | ✗ | Classification |
| Split-CIFAR-100 (Zenke et al., 2017) | 50K | 2008 | ✗ | ✗ | Classification |
| Split-MINI-ImageNet (Chaudhry et al., 2019) | 50K | 2009 | ✗ | ✗ | Classification |
| Split-ImageNet (Wen et al., 2020) | 1.2M | 2009 | ✗ | ✗ | Classification |
| Split-ImageNet-R (Wang et al., 2022) | 30K | 2019 | ✗ | ✗ | Classification |
| CORe50 (Lomonaco & Maltoni, 2017) | 165K | 2017 | ✗ | ✗ | Detection |
| CLAD (Verwimp et al., 2023) | 23K | 2021 | ✗ | ✗ | Detection |
| WANDERLUST (Wang et al., 2021) | 326K | 2021 | ✓ | ✗ | Detection |
| Inc-PASCAL (Michieli & Zanuttigh, 2019) | 11K | 2012 | ✗ | ✗ | Segmentation |
| Inc-ADE20K (Cermelli et al., 2020) | 20K | 2012 | ✗ | ✗ | Segmentation |
| StreamingQA (Liška et al., 2022) | 100K | 2007–2020 | ✓ | ✗ | Question Answering |
| TemporalWiki (Jang et al., 2022) | 32M | 2021 | ✓ | ✗ | Language Modeling |
| CKL (Jang et al., 2021) | 30K | 2019-2021 | ✗ | ✗ | Language Modeling |
| CTrL (Veniat et al., 2020) | 300K | 1998-2017 | ✗ | ✗ | Classification |
| CLOC (Cai et al., 2021) | 39M | 2006-2014 | ✓ | ✗ | Classification |
| CLEAR (Lin et al., 2021) | 7.8M | 2004-2014 | ✓ | ✗ | Classification |
| NEVIS (Bornschein et al., 2022) | 8M | 1992-2021 | ✓ | ✗ | Classification |
| Mod-X (Ni et al., 2023) | 156K | 2014 | ✗ | ✓ | Retrieval |
| CLiMB (Srinivasan et al., 2022) | 1.3M | 2013-2021 | ✗ | ✓ | Classification |
| TIC-YFCC | 15M | 2008-2014 | ✓ | ✓ | Retrieval / ZS Classification |
| TIC-RedCaps | 12M | 2011-2020 | ✓ | ✓ | Retrieval / ZS Classification |
| TIC-DataComp | 100M/1B/12B | 2014-2022 | ✓ | ✓ | Retrieval / ZS Classification |

## A.1  EXTENDED RELATED WORK

Neural networks trained on new data suffer from catastrophic forgetting of prior knowledge (Sutton, 1986; Goodfellow et al., 2013). Addressing the continual learning challenge, researchers have primarily honed in on methods tailored for small-scale benchmarks, specifically focusing on domain, class, or task incremental benchmarks (Hsu et al., 2018; Van de Ven & Tolias, 2019). Continual learning of foundation models would significantly reduce the costs and increase quick adaptability. While some recent works have started to introduce continual learning benchmarks, they are not naturally time-continual and are comparatively much smaller in scale (Ni et al., 2023; Srinivasan et al., 2022). While evaluations on these benchmarks often neglect the consideration of "training time", it becomes a pivotal factor when scaling continual learning approaches to scenarios involving the training of foundation models such as CLIP.

In our study, we abstain from comparing with continual learning methods that notably prolong the "training time". Methods such as GEM (Lopez-Paz & Ranzato, 2017; Chaudhry et al., 2018), and IMM (Lee et al., 2017), which compute gradients for two models in each training iteration, essentially double the training duration. For completeness, we include a comparison with LWF (Li & Hoiem, 2017; Ding et al., 2022) and EWC (Kirkpatrick et al., 2017). While these methods increase computation cost over standard training due to an additional forward pass, the increase in computation cost is relatively much smaller than methods that compute additional gradients. Our LWF implementation is motivated by Ding et al. (2022) which focuses on continual fine-tuning CLIP models on classification tasks by adapting LwF to CLIP models. Instead, for setups where additional compute resources are available, we run our Cumulative-All approach for slightly longer. Cumulative-All narrows the gap with Oracle (refer to Table 2). Given that data storage costs are substantially lower than computational costs at scale, we advocate for taking computational efficiency into consideration in future endeavors.

## A.2  DISCUSSION AND COMPARISON WITH CLOC BENCHMARK

Cai et al. (2021) provide interesting discussion/analysis for continual learning at a large number of steps. However, our study differs from Cai et al. (2021) in several crucial respects: (i) Training

Methodology: We employ noisy supervision using contrastive loss between image-text pairs, as opposed to the cross-entropy loss used by Cai et al. (2021). (ii) Scale of Experiments: Our experiments on the TiC-DataComp dataset are orders of magnitude larger, scaling up by $200\times$.

These differences introduce unique challenges. The use of contrastive loss (i) necessitates a tailored approach to designing our evaluation studies. The significantly larger scale of our experiments (ii) poses challenges in collecting timestamped data and understanding if and how distribution shifts impact learning at this scale.

# B    ADDITIONAL EXPERIMENTAL RESULTS

## B.1    DETAILED RESULTS ON OUR BENCHMARKS

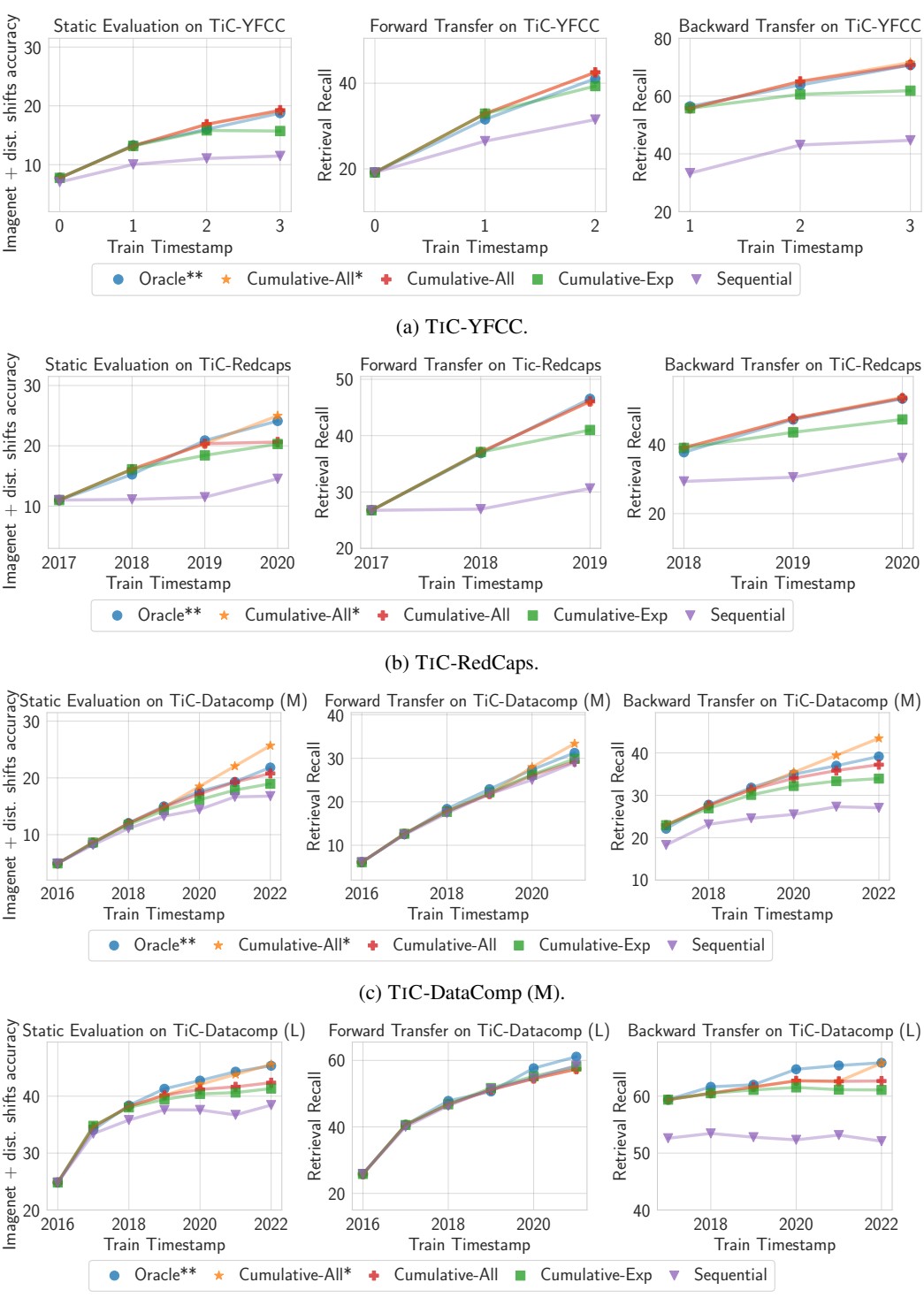

(a) TiC-YFCC.

(b) TiC-RedCaps.

(c) TiC-DataComp (M).

(d) TiC-DataComp (L).

Figure 5: **Static and dynamic evaluation performance over time with selected methods in our testbed.** As we get more data, all methods improve on both static and forward transfer on dynamic tasks but methods with limited replay buffer start performing slightly worse for backward transfer.

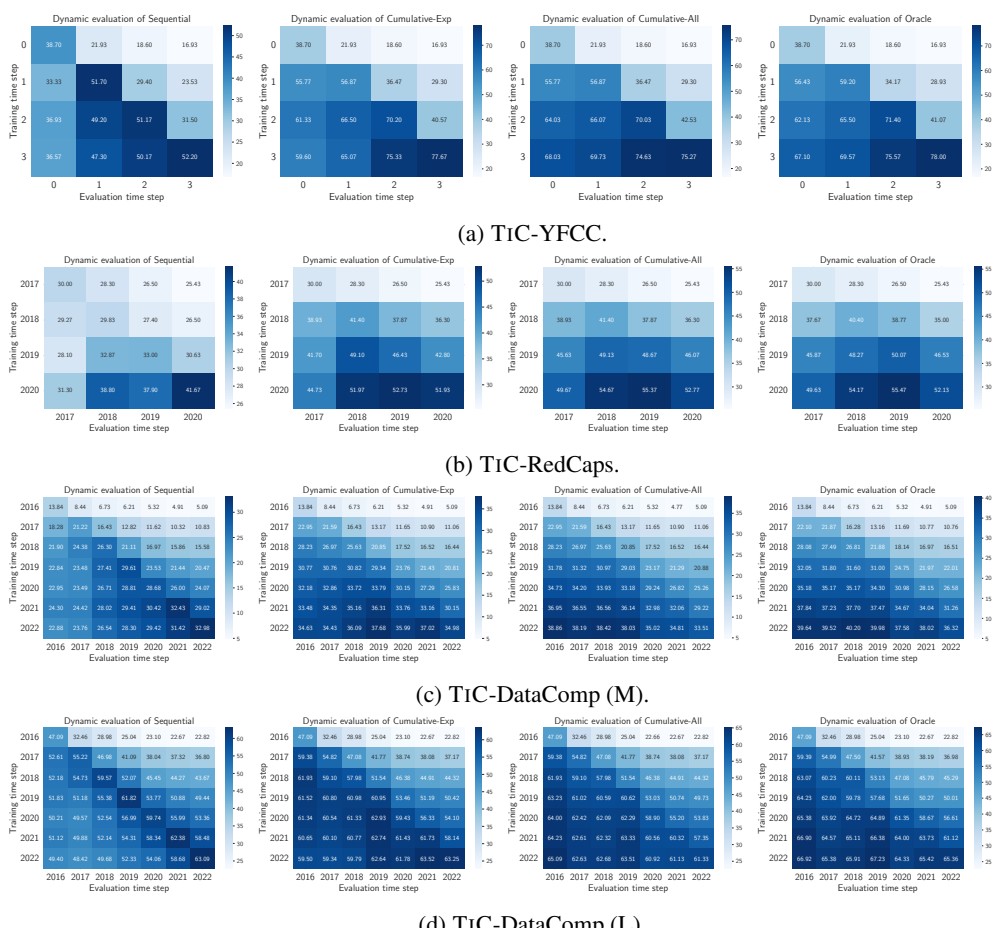

(a) TIC-YFCC.

(b) TIC-RedCaps.

(c) TIC-DataComp (M).

(d) TIC-DataComp (L).

Figure 6: Dynamic retrieval evaluation results on our benchmarks with Sequential, Cumulative-Exp, Cumulative-All and Oracle. These evaluations highlight the catastrophic forgetting observed with Sequential and Cumulative-Exp. Moreover, by observing new data, we not only benefit on tasks from current time step but also improve performance on tasks from old time steps.

## B.2    RESULTS WITH BASIC FILTERING ON TIC-DATACOMP XL

**Filtering strategy changes the ordering of performance on static and dynamic retrieval tasks.** We observe that while Bestpool filtering models outperform basic filterining models on TIC-DataComp (XL) by 6% on static tasks, they underperform by over $5\%$ on dynamic retrieval task (see Fig. 7). In the main paper (Table 2), we included TIC-DataComp (xlarge) results with Bestpool filtering. In Table 5, we include basic filtering results. We observe that while Bestpool filtering models perform better than basic filtering models on static tasks, the order is flipped on dynamic retrieval tasks. Hence, we resort to including results with Basic filtering at smaller scales, but include Bestpool results for completeness as it achieves better results on static tasks.

Table 5: **Zero shot performance on our time-continual benchmarks (Basic and Bestpool filtering).**
* and ** denote methods that violate the compute budget and use extra compute. For static tasks, we tabulate accuracy of the models obtained on the final timestamp. For dynamic tasks, we tabulate forward transfer, backward transfer and ID performance. For all metrics, higher is better. Bestpool filtering results are copied from Table 2.

| Benchmark | Method | Compute (MACs) | Static Tasks | | | | Dynamic Retrieval Tasks | | |
|---|---|---|---|---|---|---|---|---|---|
| | | | ImageNet | ImageNet dist. shift | Flickr30k | Average over 28 datasets | Backward Transfer | ID Perfor-mance | Forward Transfer |
| **TıC-DataComp** | Sequential | $2.7 \times 10^{20}$ | 66.5 | 54.2 | 61.2 | 61.0 | 63.1 | 68.9 | 56.8 |
| (XL; Bestpool) | Cumulative-All | $2.7 \times 10^{20}$ | 71.6 | 58.8 | 65.1 | 64.8 | **70.7** | **68.5** | **57.1** |
| | Cumulative-All* | $3.5 \times 10^{20}$ | **72.8** | 60.4 | 66.5 | **66.7** | 71.0 | 68.6 | 57.1 |
| | Oracle** | $1.1 \times 10^{21}$ | **73.3** | **61.3** | **68.0** | 65.8 | - | - | - |
| **TıC-DataComp** | Cumulative-All | $2.7 \times 10^{20}$ | 63.5 | 52.0 | 62.8 | 58.7 | 64.6 | 55.5 | 47.6 |
| (XL; Basic) | Sequential | $2.7 \times 10^{20}$ | 60.2 | 48.9 | 62.4 | 56.6 | 51.6 | 50.3 | 45.0 |
| | Oracle** | $1.1 \times 10^{21}$ | 66.0 | 54.0 | 63.8 | 59.6 | - | - | - |

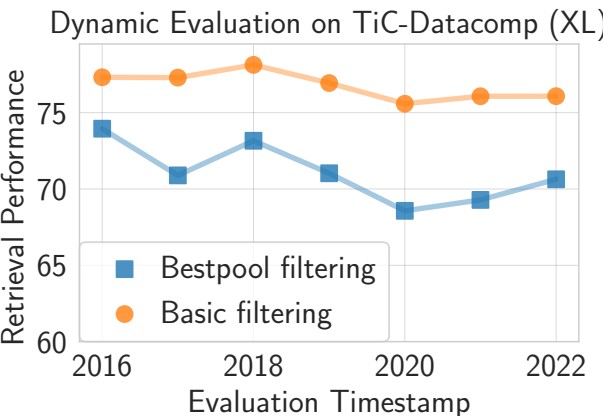

Figure 7: Comparing Oracle models trained on Bestpool and Basic filtering trained on data from all time steps. Our results clearly highlight that Basic filtering performs better than Bestpool filtering on dynamic retrieval task. However, on static tasks, the order is reversed. Moreover, Bestpool filtering shows a drop in retrieval performance from 2016 to 2022 when compared with Basic filtering.

### B.3 ABLATIONS WITH LEARNING RATE WARMUP AND MAXIMUM LEARNING RATE

To continually train models as more data arrives sequentially over time, we use multiple cycles of cosine learning rate schedule (Fig. 8). There are two crucial design choices: (i) Should we warm up the learning rate for subsequent continual runs? and (ii) How should the maximum learning rate change for sequential training runs?

Table 6: **Zero shot performance on our time-continual benchmarks with and without initial LR wamrup for subsequent runs.** Using warm up on sequential runs after training on the first time step hurts slightly when compared with not using warm up on sequential runs.

| Benchmark | Method | Static Tasks | | | | Dynamic Retrieval Tasks | | |
|---|---|---|---|---|---|---|---|---|
| | | ImageNet | ImageNet dist. shift | Flickr30k | Average over 28 datasets | Backward Transfer | ID Perfor-mance | Forward Transfer |
| **TıC-DataComp** (M) | Cumulative-All (w/o warmup) | 24.0 | 20.2 | 20.9 | 17.9 | 33.8 | 26.4 | 15.1 |
| | Cumulative-All (w warmup) | 23.3 | 20.1 | 20.3 | 17.6 | 33.3 | 26.1 | 14.8 |
| **TıC-DataComp** (L) | Cumulative-All (w/o warmup) | 48.9 | 41.3 | 50.9 | 36.3 | 62.1 | 57.3 | 41.2 |
| | Cumulative-All (w warmup) | 47.6 | 40.6 | 50.0 | 35.2 | 60.1 | 53.0 | 39.5 |

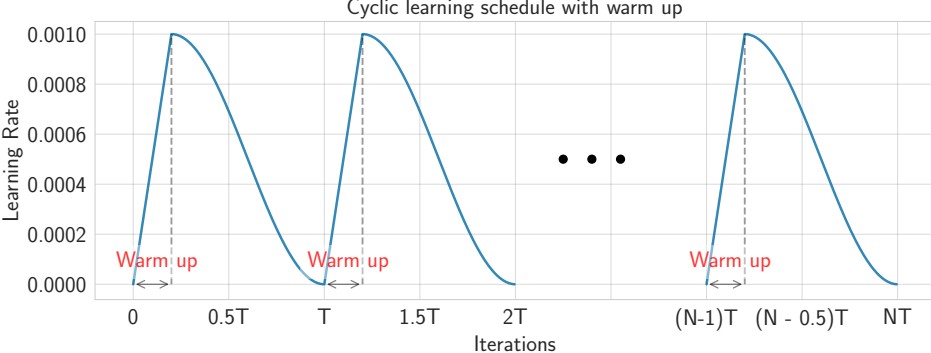

(a) Multiple cycles of standard cosine learning rate schedules which involves warm-up for all subsequent training runs.

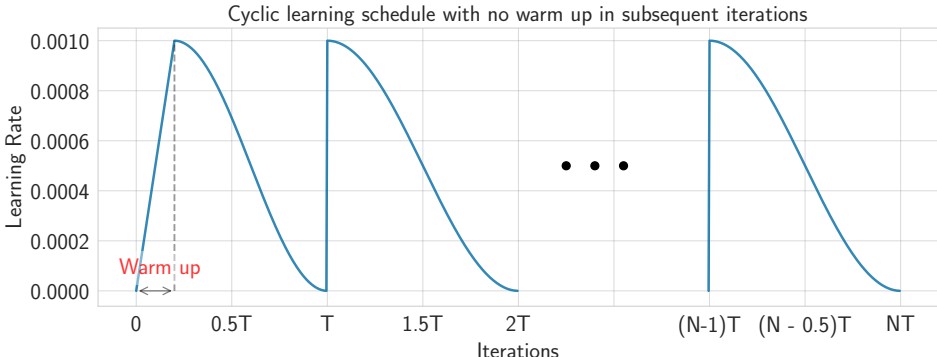

(b) Our proposed cosine learning rate schedule without learning rate warm-up for subsequent training runs.

Figure 8: **Learning rate schedule ablations.** Schedules vary on how continual training is performed when the training run is initialized with the best previous model. When training with cosine learning schedules for subsequent runs, we observe that keeping the same maximum learning rate as the first run performs the best.

Table 7: Cumulative experiments on TIC-DataComp (M) with different maximum learning rates for subsequent runs with first run fixed at LR 0.00025. Our default choice for subsequent runs is 0.00025. Performance reported on ImageNet. At maximum learning rate 0.001, the runs crashed with Nan in loss.

| Method | Max LR | | | | |
|---|---|---|---|---|---|
| | 0.00005 | 0.0001 | 0.00025 | 0.0005 | 0.001 |
| Cumulative-All | 16.3 | 19.0 | 24.0 | 10.1 | – |

When training with large batches, linear learning rate warm-up is typically employed to stabilize the start of the training when beginning from a random initialization (Goyal et al., 2017; Steiner et al., 2021). However, when training sequentially by initializing models with checkpoints from the previous step, it remains unclear whether we should employ a learning rate warm up or not. Our observations highlight that while warm up is benefits for the first time step, not using warm up on subsequent runs performs better. In particular, we observe that removing the warm up for the first training run hurts the final performance. On TIC-DataComp (`large`), we observe that training a ViT-B/16 with warm up on the first time step (i.e., 2016) gets 29.9 zero-shot on Imagenet, whereas, without warm up ViT-B/16 achieves only 24.1 zero-shot performance on Imagenet. Table 6 shows the final performance of models trained with and without warmup on subsequent time steps (after training on the first time step with

warmup). In particular, on TIC-DataComp (`large`), we observe $1.5\%$ accuracy gap on Imagenet and $4.3\%$ accuracy gap on dynamic ID retrieval performance on models trained with and without warm up.

Hence, we default to using warmup when training on the first time step and not using it on the subsequent time steps with all methods except for training on TIC-DataComp (XL) where we add a smaller warm up (10% of the warm up iterations used in first step) to stabilize training.

Next, we experiment with different maximum learning rate when training with cosine schedules. We ablate on TIC-DataComp (M) to investigate how to change LR after training on data from the first time step. Unlike conventional pretraining and finetuning settings where LR is typically decreased for subsequent training, we observe that decaying maximum LR for subsequent steps in our setup hurts on static and dynamic tasks and consequently, we use the same maximum LR across our runs (see Table 7).

### B.4 PRELIMINARY EXPERIMENTS COMPARING RANDOM SUBSAMPLING WITH OTHER STRATEGIES TO REDUCE BUFFER SIZE

In our preliminary experiments, we explored the efficacy of subsampling old data based on the alignment between text and image content from previous time steps. Specifically, when training a model at time step $t + 1$, we used the model from the end of time step t to assess this alignment. We employed two distinct subsampling methods:

1. Retaining half of the data with the lowest alignment scores, based on the premise that these data points might be more challenging to learn and require additional gradient steps.

2. Retaining half of the data with the highest alignment scores, under the assumption that these represent higher quality data, as indicated by the stronger alignment between text and image pairs.

We applied these methods to the TiC-YFCC dataset and evaluated their performance against a baseline of random sampling. The outcomes revealed minimal differences: less than 0.2% variation in Imagenet performance and under 0.5% in dynamic retrieval performance across different time steps. Given that these minor improvements came with a significant computational cost—requiring a full forward pass to compute alignment post each training epoch—they exceeded our compute budget constraints. As a result, we opted for random sampling in our research. We leave investigation on improved subsampling techniques for future work.

### B.5 CONST-COSINE: AN ALTERNATIVE LEARNING RATE SCHEDULE

The defacto LR schedule for training CLIP models is an initial linear increase to a maximum value, i.e., warm up, followed by a cosine decay (Radford et al., 2021; Gadre et al., 2023). In the main paper, we default to using cosine LR schedule for each sequential run, resulting in a cyclic schedule. We observe a significant increase in training loss early in subsequent runs when the LR is high. Comparing the loss on training data with Cumulative and Oracle methods, we observe that as training progresses the training loss increases every time the learning rate is increased to the maximum LR (Fig. 9).

It would be ideal for continual training to employ a learning rate schedule that is "forward looking", allowing us to continually train from a previous checkpoint without experiencing a significant increase in training loss. One desirable property of such a learning rate schedule would be its ability to adapt without requiring prior knowledge of the decay period.

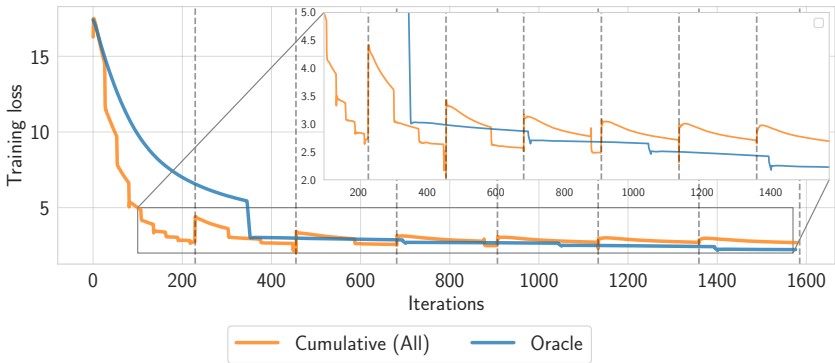

Figure 9: **Training loss increases every time the LR is reset to maximum LR for Cumulative.** Loss comparison on training data with Cumulative and Oracle method. Cumulative is trained with a cyclic cosine schedule without warm up for sequential training runs. For Cumulative, we plot the loss on training data, and as the training progresses, samples from new time steps are added to the training pool. For Oracle, the training data is the union of data from all time steps and remains the same throughout the training.

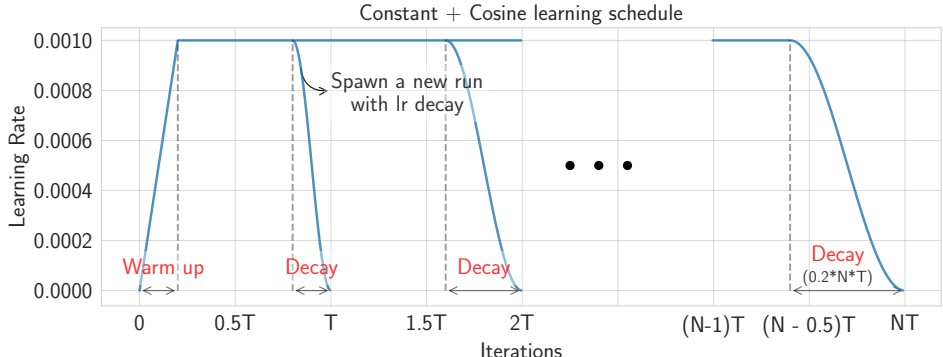

Figure 10: Const-Cosine: Our proposed alternative forward-looking learning rate schedule schedule which trains one model with constant learning rate and decays the learning rate with cosine schedule only for a fraction of iterations before obtaining a deployable model. Const-Cosine schedule uses an extra compute budget than an Oracle run because an extra training run is launched for the fraction of training when learning rate is decayed.

In our work, we perform preliminary experiments with the *simplest* alternative, Const-Cosine where after the warm up period, we train with a constant learning rate and decay the learning rate only for a small fraction of training towards the end when we want a deployable model (Fig. 10). This allows us to continue training for subsequent runs from the checkpoint at the end of the constant learning rate schedule and decay the LR only in the end. For our experiments, we fix the decay period as $0.2$ of the total training iterations. Due to this, Const-Cosine schedule slightly increases the overall training budget of the Cumulative runs when compared with cyclic cosine schedules.

For Const-Cosine, we only ablate at relatively smaller scale datasets in our testbed (i.e., TIC-YFCC, TIC-RedCaps, and TIC-DataComp (`medium`)). For a fair comparison, we also re-run Oracle methods with the same Const-Cosine schedule. Note that for Const-Cosine experiments, we use the same maximum LR as with the cosine schedule.

We observe that training with Const-Cosine schedule significantly improves both Cumulative and Oracle as compared to their counterparts trained with cosine learning rates [2]. Moreover, as expected, we

---

[2]We also experimented with Const-Cosine schedule for Oracle training on TIC-DataComp (`large`) and TIC-DataComp (`xlarge`). We observe that with a decay fraction of $0.2$, Const-Cosine achieves similar results

do not observe jumps in training loss when training Cumulative with Const-Cosine schedule. However, the gap between Oracle and Cumulative with Const-Cosine doesn't decrease when compared with gap between Oracle and Cumulative with cosine learning rate schedules. This highlights that the jumps in the training loss observed while training with the cyclic cosine schedule might have benign effects on the final performance.

Table 8: **Zero shot performance on Imagenet with Const-Cosine LR schedule.** We observe that Const-Cosine improves over cyclic cosine LR schedule. However, the gap between cyclic cosine LR schedule and Const-Cosine for different LR schedules remains the same. ** denote methods that violate the compute budget.

| Benchmark | Method | Cosine LR Schedule | | Const-Cosine LR schedule | |
|---|---|---|---|---|---|
| | | Compute (MACs) | ImageNet | Compute (MACs) | ImageNet |
| TiC-YFCC | Cumulative-All | $3.4 \times 10^{18}$ | 29.3 | $4.4 \times 10^{18}$ | 32.8 |
| | Oracle** | $8.5 \times 10^{18}$ | 29.2 | $8.5 \times 10^{18}$ | 33.2 |
| TiC-RedCaps | Cumulative-All | $3.4 \times 10^{18}$ | 32.2 | $4.4 \times 10^{18}$ | 35.1 |
| | Oracle** | $8.5 \times 10^{18}$ | 32.7 | $8.5 \times 10^{18}$ | 36.2 |
| TiC-DataComp (M) | Cumulative-All | $3.0 \times 10^{18}$ | 24.0 | $3.6 \times 10^{18}$ | 28.2 |
| | Oracle** | $1.2 \times 10^{19}$ | 25.5 | $1.2 \times 10^{19}$ | 28.9 |

## B.6 OPENCLIP MODELS OBTAINED BY RETRAINING AFTER REMOVING ANY DUPLICATE EXAMPLES FROM THE TEST SET

OpenCLIP models (e.g., models trained on Datacomp and LAION-5B) have been trained on data curated from Common Crawl. Since the retrieval tasks we constructed are built on top of data curated from Common Crawl, one may argue there is a possibility of train/test overlap in our evaluations of OpenCLIP models. Thus, we retrain OpenCLIP models on DataComp datasets after removing the samples in our test sets. Figure 11 shows that the trends observed for OpenCLIP models holds for our retrained models.

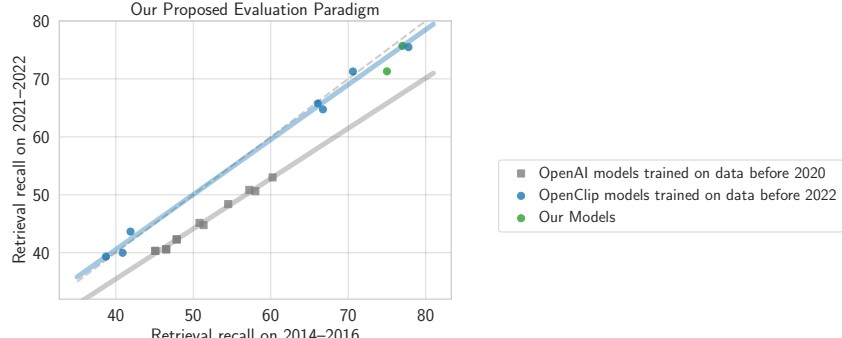

Figure 11: We replicate OpenCLIP models by training from scratch and removing duplicates from the evaluation dataset. We observe that trends continue to hold.

## B.7 RESULTS ON DYNAMIC CLASSIFICATION TASK

In the main paper, we include results on our dynamic retrieval task. For completeness, here we include results on dynamic classification tasks on TiC-DataComp splits (Table 9). Along with including results on all nodes of ImageNet, we also include results on classification task restricted to classes in the "motor vehicles" subtree of ImageNet hierarchy. For the dynamic classification task, we observe trends similar to the dynamic retrieval task.

---

to that of the cosine learning rate schedule. In particular, Const-Cosine achieves 61.3 on `large` and 73.0 on `xlarge` versus Cosine schedule achieves 62.3 on `large` and 73.3 on `xlarge`. This highlights the potential of training with Const-Cosine schedule in scenarios where total training duration might be unknown apriori.

Table 9: **Zero shot performance on our TIC-DataComp-Net classification task.** * and ** denote methods that violate the compute budget. We tabulate forward/backward transfer and ID performance on classification tasks (Sec. 2.3). For TIC-DataComp (XL), we include results with Bestpool filtering.

| Benchmark | Method | Compute (MACs) | Dynamic Retrieval Tasks (All) | | | Dynamic Retrieval Tasks ('Motor Vehicles') | | |
|---|---|---|---|---|---|---|---|---|
| | | | Backward Transfer | ID Perfor-mance | Forward Transfer | Backward Transfer | ID Perfor-mance | Forward Transfer |
| **TIC-DataComp** (M) | Sequential | $3.0 \times 10^{18}$ | 15.9 | 13.3 | 9.9 | 34.5 | 30.0 | 22.6 |
| | Patching | $3.0 \times 10^{18}$ | 15.6 | 13.1 | 9.7 | 34.4 | 29.2 | 22.1 |
| | Cumulative-Exp | $3.0 \times 10^{18}$ | 17.6 | 14.4 | 10.4 | 36.6 | 30.9 | 23.5 |
| | Cumulative-Equal | $3.0 \times 10^{18}$ | 17.5 | 14.2 | 10.4 | 36.4 | 31.1 | 23.5 |
| | Cumulative-All | $3.0 \times 10^{18}$ | 18.3 | 14.7 | 10.6 | 38.2 | 31.7 | 23.7 |
| | LwF* | $3.8 \times 10^{18}$ | 16.0 | 13.5 | 9.9 | 35.1 | 30.7 | 23.3 |
| | Cumulative-All* | $3.9 \times 10^{18}$ | 20.7 | 16.0 | 10.9 | 40.4 | 32.3 | 23.9 |
| | Oracle** | $1.2 \times 10^{19}$ | 19.2 | 15.2 | 10.7 | 38.7 | 31.9 | 23.5 |
| **TIC-DataComp** (L) | Sequential | $2.7 \times 10^{19}$ | 38.3 | 36.9 | 33.3 | 58.4 | 55.6 | 49.7 |
| | Patching | $2.7 \times 10^{19}$ | 38.6 | 36.8 | 33.3 | 58.3 | 54.9 | 49.3 |
| | Cumulative-Exp | $2.7 \times 10^{19}$ | 40.2 | 37.9 | 34.2 | 60.7 | 56.8 | 51.1 |
| | Cumulative-Equal | $2.7 \times 10^{19}$ | 40.6 | 38.0 | 34.2 | 60.7 | 56.8 | 50.8 |
| | Cumulative-All | $2.7 \times 10^{19}$ | 41.3 | 38.3 | 34.4 | 61.4 | 56.6 | 50.9 |
| | Cumulative-All* | $4.1 \times 10^{19}$ | 43.0 | 39.2 | 34.6 | 62.7 | 57.5 | 51.1 |
| | Oracle** | $1.1 \times 10^{20}$ | 43.8 | 40.0 | 35.2 | 62.6 | 56.8 | 50.7 |
| **TIC-DataComp** (XL) | Sequential | $2.7 \times 10^{20}$ | 55.4 | 55.1 | 53.3 | 67.8 | 66.0 | 63.5 |
| | Cumulative-All | $2.7 \times 10^{20}$ | 58.5 | 56.7 | 54.3 | 70.2 | 67.4 | 63.8 |
| | Cumulative-All* | $3.5 \times 10^{20}$ | 58.8 | 56.9 | 54.3 | 70.5 | 67.5 | 63.8 |

## B.8 ADDRESSING DIFFERENCES BETWEEN SEQUENTIAL AND CUMULATIVE-ALL BETWEEN TIC-YFCC AND TIC-DATACOMP

In Table 2, we observe differences in the behavior of Sequential and Cumulative-Allon TIC-YFCC when compared with TIC-DataComp. For instance, differences between the ID performance between Sequential and Cumulative-All is larger in TIC-YFCC than in TIC-DataComp (M). Similar observations hold true for backward transfer performance. In this section, we explain the underlying causes for these differences.

We identify two primary reasons:

(i) the nature of the distribution shift observed in TIC-YFCC. We observe that models trained with Sequential on TIC-YFCC suffer from relatively larger drops on old-time steps than TIC-DataComp (M) due to catastrophic forgetting (see Fig. 6).

(ii) compute used at each time step per data available at each time step is different for these bencmarks. Overall YFCC is 2x smaller than Tic-Datacomp (M) but the compute we used in both TiC-YFCC and TiC-Datacomp setup is of similar order (in fact, it is slightly higher in TiC-YFCC). We re-ran the experiments for Tic-YFCC by reducing the compute. In the updated runs, we observe that the gap between ID performances of Sequential and Cumulative-All vanishes.

Table 10: **Zero shot retrieval performance on TIC-YFCC with Sequential and Cumulative-All with reduced compute.**

| Benchmark | Method | Dynamic Retrieval Tasks (original compute) | | | | Dynamic Retrieval Tasks (reduced compute) | | | |
|---|---|---|---|---|---|---|---|---|---|
| | | Compute (MACs) | Backward Transfer | ID Perfor-mance | Forward Transfer | Compute (MACs) | Backward Transfer | ID Perfor-mance | Forward Transfer |
| **TIC-YFCC** | Sequential | $3.4 \times 10^{18}$ | 42.2 | 48.4 | 23.7 | $1.5 \times 10^{18}$ | 27.0 | 42.0 | 15.7 |
| | Cumulative-All | $3.4 \times 10^{18}$ | 66.4 | 60.2 | 27.6 | $1.5 \times 10^{18}$ | 46.3 | 38.7 | 17.3 |

## C ADDITIONAL BENCHMARK DETAILS

### C.1 FILTERING ABLATIONS ON TIC-DATACOMP

For Basic Filtering, Gadre et al. (2023) performs the following three steps: filter by English language (using fasttext (Joulin et al., 2017)), filter by caption length over two words and 5 characters, and filter by image sizes with smallest dimensions over 200 pixels and aspect ratio above 3. We do not default to other filtering techniques that use off-the-shelf CLIP models from Gadre et al. (2023) to

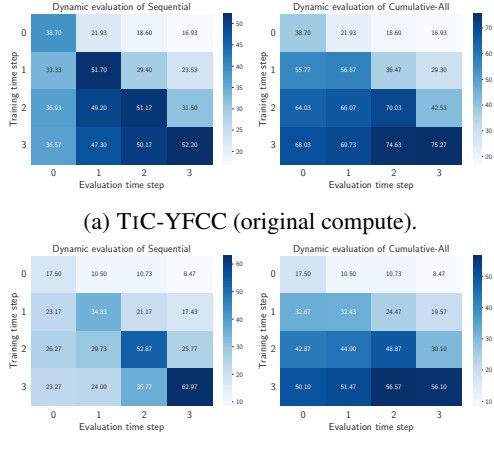

(a) TIC-YFCC (original compute).

(b) TIC-YFCC (reduced compute).

Figure 12: Dynamic retrieval evaluation results with Sequential, Cumulative-All on TIC-YFCC with reduced compute.

avoid biasing dataset selection from each time step. In Fig. 13, we show that "Bestpool" filtering (which filters image-text pairs with CLIP scores and ImageNet image embeddings) biases dataset selection to preferring old time step data over new timestamp data. Moreover, we also show that models trained with Bestpool filtering is less robust when evaluated on our dynamic tasks from 2021-2022 (Fig. 13). Nevertheless, for completeness and to highlight the significance of our findings even for state-of-the-art filtering techniques, we perform continual learning experiments with Bestpool filtering at `xlarge` scale which is included in the main paper. In App. B.2, we include results with Basic filtering at `xlarge`.

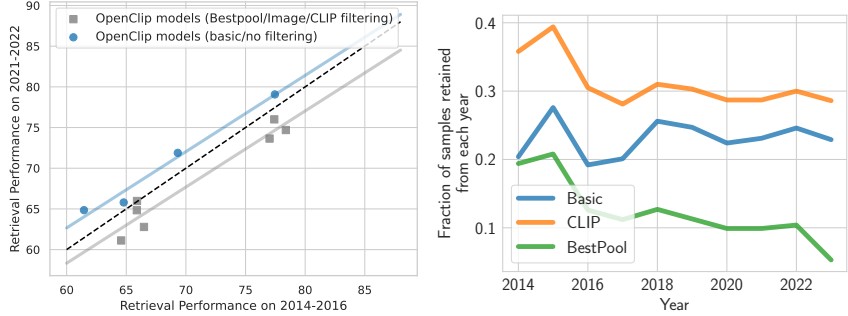

Figure 13: (Left) Gap in retrieval performance for different OpenCLIP models that use different filtering techniques. (Right) Reduction in TIC-DataComp data at different times with different filtering techniques. This clearly highlights that there is a selection bias towards retaining more old data for CLIP/BestPool filtering. No such bias exists for basic filtering.

## C.2 STATIC DATASETS CONSIDERED FOR EVALUATION

Table 11: Evaluation tasks borrowed from Gadre et al. (2023).

| Task type | Dataset | Task | Test set size | Number of classes | Main metric |
|---|---|---|---|---|---|
| Classification | Food-101 Bossard et al. (2014) | Food recognition | 25,250 | 101 | accuracy |
| | GTSRB Stallkamp et al. (2011) | Traffic sign recognition | 12,630 | 43 | accuracy |
| | ImageNet 1k Deng et al. (2009) | Visual recognition | 50,000 | 1,000 | accuracy |
| | ImageNet Sketch Wang et al. (2019) | Visual recognition | 50,889 | 1,000 | accuracy |
| | ImageNet V2 Recht et al. (2019) | Visual recognition | 10,000 | 1,000 | accuracy |
| | ImageNet-A Hendrycks et al. (2021b) | Visual recognition | 7,500 | 200 | accuracy |
| | ImageNet-O Hendrycks et al. (2021b) | Visual recognition | 2,000 | 200 | accuracy |
| | ImageNet-R Hendrycks et al. (2021a) | Visual recognition | 30,000 | 200 | accuracy |
| | KITTI distance Geiger et al. (2012); Zhai et al. (2019) | Distance prediction | 711 | 4 | accuracy |
| | MNIST LeCun (1998) | Digit recognition | 10,000 | 10 | accuracy |
| | ObjectNet Barbu et al. (2019) | Visual recognition | 18,574 | 113 | accuracy |
| | Oxford Flowers-102 Nilsback & Zisserman (2008) | Flower recognition | 6,149 | 102 | mean per class |
| | Oxford-IIIT Pet Parkhi et al. (2012); Zhai et al. (2019) | Pet classification | 3,669 | 37 | mean per class |
| | Pascal VOC 2007 Everingham et al. (2007) | Object recognition | 14,976 | 20 | accuracy |
| | PatchCamelyon Veeling et al. (2018); Zhai et al. (2019) | Metastatic tissue cls. | 32,768 | 2 | accuracy |
| | Rendered SST2 Zhai et al. (2019) | Sentiment classification | 1,821 | 2 | accuracy |
| | RESISC45 Cheng et al. (2017); Zhai et al. (2019) | Satellite imagery recognition | 6,300 | 45 | accuracy |
| | Stanford Cars Krause et al. (2013) | Vehicle recognition | 8,041 | 196 | accuracy |
| | STL-10 Coates et al. (2011) | Visual recognition | 8,000 | 10 | accuracy |
| | SUN-397 Xiao et al. (2016) | Scene recognition | 108,754 | 397 | accuracy |
| | SVHN Netzer et al. (2011); Zhai et al. (2019) | Digit recognition | 26032 | 10 | accuracy |
| | iWildCam Beery et al. (2020); Koh et al. (2021) | Animal recognition | 42,791 | 182 | macro F1 score |
| | Camelyon17 Bandi et al. (2018); Koh et al. (2021) | Metastatic tissue cls. | 85,054 | 2 | accuracy |
| | FMoW Christie et al. (2018); Koh et al. (2021) | Satellite imagery recognition | 22,108 | 62 | worst-region acc. |
| Retrieval | Flickr30k Young et al. (2014) | Image and text retrieval | 31,014 | N/A | R@1 |

## C.3 OUR BENCHMARK STATISTICS

In this section, we discuss statistics of our constructed benchmarks. Fig. 14 summarizes TIC-RedCaps, TIC-YFCC and TIC-DataComp dataset sizes. Fig. 15 summarizes original YFCC dataset sizes. Table 12, Table 13 and Table 14 present the exact numbers for these datasets. For TIC-DataComp, we only discuss the sizes at `xlarge` scale.

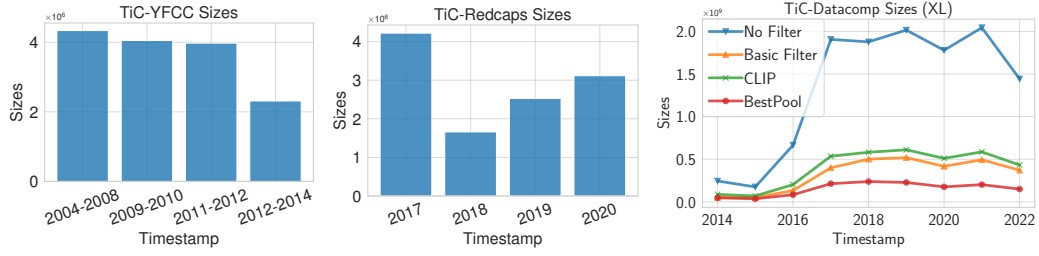

Figure 14: Number of examples in each year in our benchmarks.

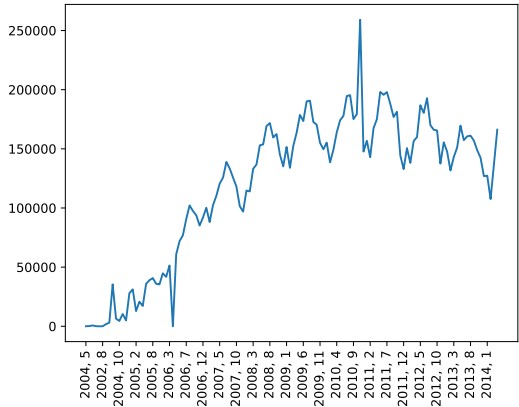

Figure 15: Number of examples in each year in original YFCC 15M. X-axis the upload month and y-axis is the number of examples in that month.

Table 12: Number of examples in TIC-RedCaps in each year.

| Dataset | Year | | | |
|---|---|---|---|---|
| | 2017 | 2018 | 2019 | 2020 |
| TIC-RedCaps | 4,220,262 | 1,660,003 | 2,526,575 | 3,115,715 |

Table 13: Number of examples in TIC-YFCC in each year.

| Dataset | Year | | | |
|---|---|---|---|---|
| | 2004–2008 | 2009–2010 | 2011–2012 | 2012–2014 |
| TIC-YFCC | 4,337,727 | 4,050,166 | 3,976,339 | 2,312,753 |

Table 14: Number of examples in TIC-DataComp in each year before filtering.

| Dataset | Year | | | | | | | | |
|---|---|---|---|---|---|---|---|---|---|
| | 2014 | 2015 | 2016 | 2017 | 2018 | 2019 | 2020 | 2021 | 2022 |
| TIC-DataComp (no filter) | 244,802,598 | 175,648,045 | 666,019,511 | 1,906,357,755 | 1,877,561,875 | 2,016,011,588 | 1,778,751,066 | 2,044,463,701 | 1,442,233,121 |
| TIC-DataComp (basic filter) | 52,764,775 | 50,757,898 | 133,333,267 | 400,225,598 | 501,347,511 | 519,575,760 | 417,067,014 | 494,038,122 | 371,748,613 |

Next, we tabulate the number of examples in our retrieval evaluation datasets. Since the evaluation dataset sizes are different at different time steps, we subsample the dataset to a fixed size before performing retrieval evaluations. On TIC-YFCC and TIC-RedCaps, we randomly sampled 1000 image-text pairs from these evaluation datasets. For TIC-DataComp, we randomly sample 4000 image-text pairs. We repeat this process for 3 seeds and report the aggregated performance.

Table 15: Number of retrieval evaluation examples in TIC-RedCaps in each year.

| Dataset | Year | | | |
|---|---|---|---|---|
| | 2017 | 2018 | 2019 | 2020 |
| TIC-RedCaps | 31,316 | 42,539 | 16,738 | 25,565 |

Table 16: Number of retrieval evaluation examples in TIC-YFCC in each year.

| Dataset | Year | | | |
|---|---|---|---|---|
| | 2004–2008 | 2009–2010 | 2011–2012 | 2012–2014 |
| TIC-YFCC | 43,820 | 40,909 | 40,165 | 23,354 |

Table 17: Number of retrieval evaluation examples in TIC-DataComp in each year before filtering.

| Dataset | Year | | | | | | |
|---|---|---|---|---|---|---|---|
| | 2016 | 2017 | 2018 | 2019 | 2020 | 2021 | 2022 |
| TIC-DataComp | 23,085 | 39,289 | 50,450 | 53058 | 42,239 | 49,841 | 38,051 |

## C.4 Compute Constraints for Different Datasets

We closely follow compute budget constraints from Gadre et al. (2023). In particular, on TIC-DataComp, we restrict to using exactly the same amount of overall compute as fixed in Gadre et al. (2023). Below we list exact total MACs on each dataset:

- TIC-YFCC: Total MACs: $3.4 \times 10^{18}$
- TIC-RedCaps: Total MACs: $3.4 \times 10^{18}$
- TIC-DataComp `medium`: Total MACs: $3.0 \times 10^{18}$
- TIC-DataComp `large`: Total MACs: $2.7 \times 10^{19}$
- TIC-DataComp `xlarge`: Total MACs: $2.7 \times 10^{20}$

For a ViT-B architecure, these values correspond to 20k iterations on TIC-YFCC (batch size: 8192), TIC-RedCaps (batch size: 8192), 35k iterations on TIC-DataComp (M) (batch size: 4096), 157k iterations on TIC-DataComp (L) (batch size: 8192), and 143.5k iterations on TIC-DataComp (XL) (batch size: 90100). We divide these iterations equally among all time steps.

## C.5 Creation Pipeline for Evaluation Datasets

**TIC-DataComp-Retrieval** To create a retrieval task, we sample a batch of IID image-text pairs from different timestamps and evaluate text retrieval performance given the corresponding image (similarly, image retrieval given the corresponding text). Alongside general evaluations, we also construct datasets from specific domains, e.g., Covid-19 subset and Flickr subset. To create Covid-19, we filter the dataset to only retain pairs where the caption contains a mention of "covid". This search process restricts the data to time only after 2019. For the Flickr subset, we filter the dataset to only retain pairs where the corresponding "url" contains data from Flickr.

**TIC-DataComp-Net** We create our dynamic classification dataset TIC-DataComp-Net with ImageNet classes from the CommonPool data augmented with temporal information. Our construction process draws inspiration from the LAIONet construction process described in Shirali & Hardt (2023). In particular, we first filter examples where the corresponding caption contains one and only one of the synsets of ImageNet-1K. We also apply additional basic filtering (Gadre et al., 2023) to make sure that images are of at least 200 size in smallest dimension and the caption contains at least 2 words and 5 characters. After filtering for examples with ImageNet synsets, we only retain examples where the similarity—as evaluated by an off-the-shelf sentence embedding model (Reimers & Gurevych, 2019)—between imagenet synset definition and the caption exceeds a threshold of 0.5. The goal of this filtering step is to restrict examples with "high" alignment between caption and imagenet synset definition. This last step differs from the LAIONet construction. Crucially, unlike LAIONet, we do not filter the image-text pairs with CLIP similarity scores to avoid biasing the dataset selection process.

## C.6 Distribution Shift Analysis on Proposed benchmarks

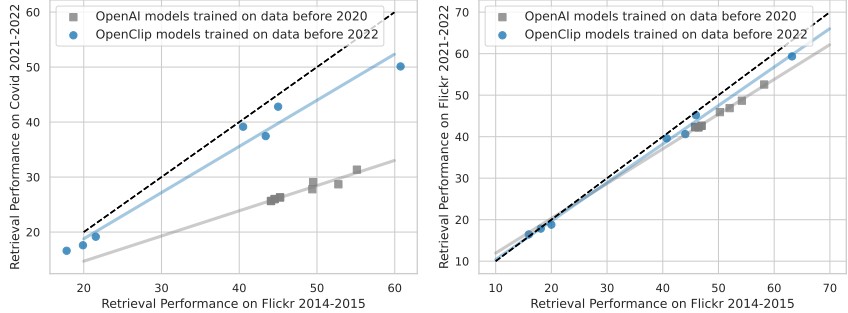

Figure 16: (Left) Comparison of retrieval performance on COVID queries versus Flickr queries (construction described in App. C.5). (Right) Comparison on old Flickr versus new Flickr data. Clearly, we observe that while gap on old versus new flickr data is small, the gap is significantly larger on Covid queries.

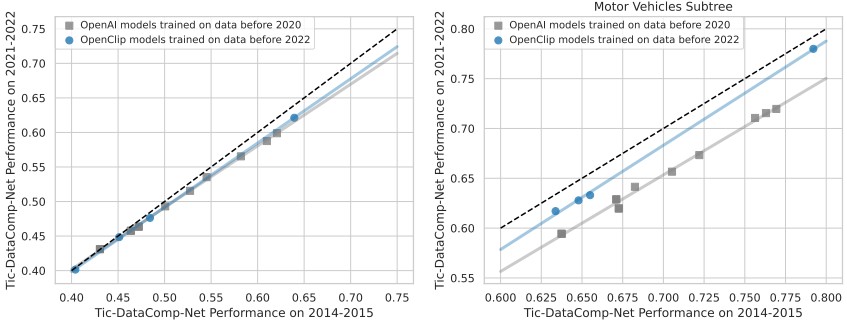

Figure 17: (Left) Comparison on old versus new data from TɪC-DataComp-Net. (Right) Comparison on motor vehicles node from TɪC-DataComp-Net. For our classification task, we observe a very small drop ($\approx 1\%$) when averaged across all categories. However, we observe a substantial gap on classes in "motor vehicle" subtree, when comparing OpenAI and OpenCLIP models. These findings highlight that while overall ImageNet classes may remain timeless, certain categories tend to evolve faster than others.

**TɪC-DataComp analysis through the lens of constructed evaluation tasks** Here, we compare performance of OpenAI and OpenCLIP models on our datasets. We observe a significant performance gap between OpenAI and OpenCLIP models on our dynamic retrieval task (Fig. 1). This gap widens notably on retrieval queries where captions mention COVID-19. On the other hand, OpenAI and OpenCLIP models exhibit similar robustness for retrieval on data coming from Flickr highlighting that data from some domains do not exhibit shifts that cause performance drops. For our classification task, we observe a very small drop ($\approx 1\%$) when averaged across all categories. However, we observe a substantial gap on specific subtrees in ImageNet. For example, classes in "motor vehicle" subtree show an approximate $7\%$ performance drop, when comparing OpenAI and OpenCLIP models. These findings highlight that while overall ImageNet classes may remain timeless, certain categories tend to evolve faster than others. Our qualitative and quantitative analysis on TɪC-DataComp clearly highlights evolution of distributions and captures different properties than standard benchmarks.

**Quantitative analysis on TɪC-YFCC** We analyze TɪC-YFCC using off-the-shelf sentence and image encoders. For off-the-shelf sentence embedder, we used an existing sentence transformer from Hugging Face (Reimers & Gurevych, 2019). For the image encoder, we use a CLIP pretrained ViT-B-16 model (Radford et al., 2021; Ilharco et al., 2021).

We first embed images from different time steps with an OpenAI CLIP encoder and then compute Frechet Inception Distance (FID; Seitzer (2020)). As time progresses, we observe that FID distance increases with respect to data from first time step (Fig. 18). Similarly, we use the pretrained sentence transformer to extract top-5 categories from Wordnet Nouns for each caption. We then obtain a distribution over these Nouns for each time step. We observe that the TV distance over the distribution of WordNet nouns evolves over time when compared to data from the first time step.

### C.7 CREATION PIPILINE FOR TɪC-DATACOMP

We collect timestamps for the CommonPool dataset introduced in DataComp. We repeat the crawling process described in Gadre et al. (2023) to download WARC files from Common Crawl. In particular, we follow the same multistep process which involved: (i) parsing URLs and alt-text from Common Crawl dumps and downloading these images; (ii) tagging images with meta data and id of the common crawl batch; and (iii) conducting evaluation set duplication and safety content filtering. After downloading the WARC files, we perform a join with the datacomp 12.8B examples. During this join, we lost approximately 0.1B of examples that are no longer available online. Moreover, while performing this join, we only retain examples with their first occurrence. This is done before running any de-duplication on image-text pairs for exact matches as done in Gadre et al. (2023).

The source of DataComp is Common Crawl, which periodically releases web-crawled data snapshots, typically on a monthly basis since 2014 with new and updated webpages. This process provides timestamps at the granularity of months, spanning years 2014–2022.

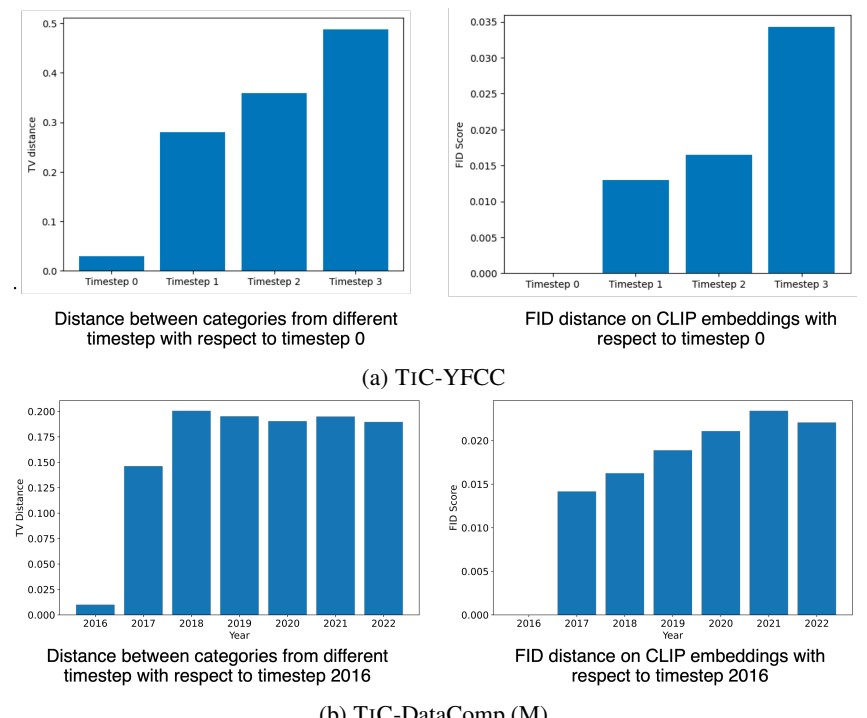

(a) TIC-YFCC

(b) TIC-DataComp (M)

Figure 18: Distribution shift results. Analysis on TIC-YFCC and TIC-DataComp (M) using off-the-shelf sentence and image encoders. We first embed images from different time steps with an OpenAI CLIP encoder and then compute Frechet Inception Distance (FID; Seitzer (2020)). As time progresses, we observe that FID distance increases with respect to data from first time step. Similarly TV distance over categorical distribution on Wordnet Noun synsets also increases with time when compared to categorical distribution on first timestep.

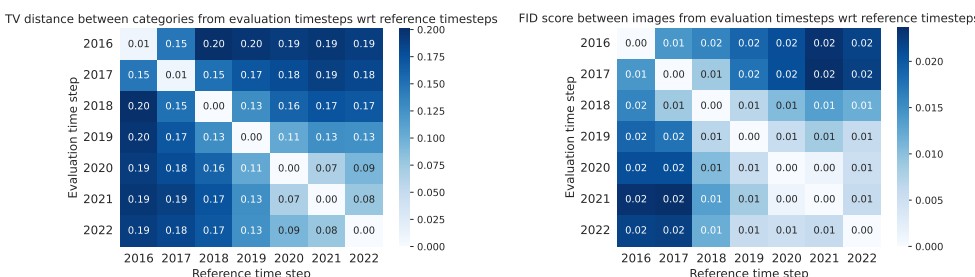

Figure 19: Distribution shift analysis on TIC-DataComp (M) using off-the-shelf sentence and image encoders. We first embed images from different time steps with an OpenAI CLIP encoder and then compute Frechet Inception Distance (FID; Seitzer (2020)). As time progresses, we observe that FID distance increases with respect to data from first time step. Similarly TV distance over categorical distribution on Wordnet Noun synsets also increases with time when compared to categorical distribution on first timestep.

We note that while this augmented time information may contain some noise, on average, we find it to be a reasonably accurate proxy for the upload time of web pages. To perform an initial check, we note that our data contains images from flickr which provides an API to query for true upload timestamp. So we extract 10k examples from our benchmark TIC-DataComp and query Flickr for their true timestamp. Fig. 20 summarizes true timestamps with timestamps extracted from CC.

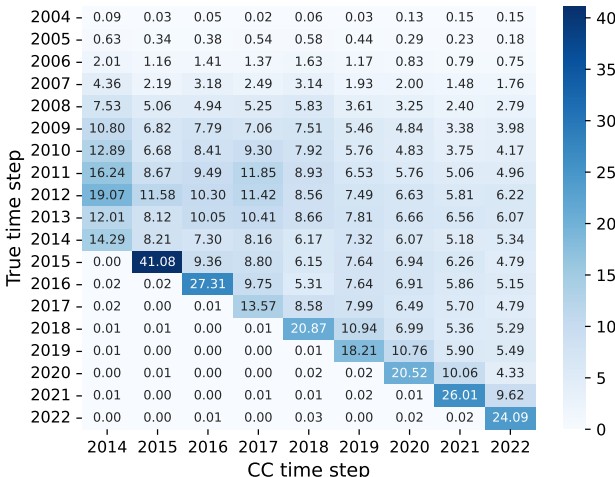

Figure 20: Comparison of Common Crawl assigned timestamp and true timestamp on a subset of 10k examples containing image-text pairs from Flickr. We observe a clear trend where CC timestamps correlate with true timestamps.

# D    ADDITIONAL EXPERIMENTAL DETAILS

## D.1    ADDITIONAL DETAILS ON IMAGENET IID SPLIT CONTINUAL LEARNING EXPERIMENT

With ImageNet data, we consider 2, 4 and 8 splits including the full dataset. This design is inspired by Ash & Adams (2020). We consider ViT-B/16 architecture trained for 300 epochs on full data and split the iterations corresponding to 300 epochs equally among k splits when training sequentially. We keep all other hyperparameters, such as learning rate, optimizer, and batch size, set to the standard values typically employed for training ViT-B/16 on the ImageNet dataset (Dosovitskiy et al., 2020). We also employ $\ell_2$ regularization and augmentation on ImageNet training data. We evaluate the models on IID ImageNet test set.

Our Imagenet experiments were primarily inspired by the "loss of plasticity" phenomenon described in Ash & Adams (2020). Their study demonstrates that models sequentially trained on two splits of CIFAR-10 data (initially on 50%, followed by 100% of data) exhibit poorer generalization compared to models trained from scratch on the entire dataset. Since we do not observe this behavior for continual training of CLIP, we investigated the existence of such behaviors on up to 8 splits of Imagenet. Our findings reveal that the simple cumulative baseline (with no extra budget) remains competitively close to the Oracle model (that benefits from using the full compute budget on the entire pooled training data from the beginning).

Prior works (Prabhu et al., 2023; Hu et al., 2021) performed continual learning experiments on Imagenet to compare different methods and highlight the effectiveness of continual training on synthetic continual learning setups derived from ImageNet. While these papers include results with an Oracle method, differences in the settings considered in these studies limit direct comparisons.

In particular, we show the performance gap of less than 1% in the same setup used otherwise in the paper when using SOTA training procedures achieving 81% validation performance. Comparitively the referenced Hu et al. (2021) does not show whether the 65% to 77% performance gap in their Table 1 can be bridged by increasing the compute for their method. Instead, authors show that if they restrict the compute for Oracle in Table 2, the Oracle performance drops to 68% (with $\approx 3\%$ gap).

Moreover, in Prabhu et al. (2023), authors perform experiments on DI-Imagenet-2k where they start with an initial memory of Imagenet-1k 1.2 M samples and sequentially observe data for the same classes 1k classes from Imagenet-21k pool. This makes comparing streaming accuracy (or Imagenet-1k accuracy) for different methods incomparable with our setup (with a gap of over 7% in streaming accuracy even at step 8 as compared to less than 1% in our setup).

## D.2 Training and Hyperparameter Details

We create a common experimental setup by fixing the training procedure for sequential runs. Unless specified otherwise, we closely follow the CLIP training recipe proposed in (Ilharco et al., 2021; Radford et al., 2021) where we train models with a contrastive objective over images and captions. Given a set of image-text pairs, we train an image encoder and a text encoder such that the similarity between the representations of images and their corresponding text is maximized relative to unaligned pairs. Only LwF deviates from this standard training procedure. For each benchmark, we pick Vision Transformers (ViTs) as the image encoder, in particular, we fix the model architecture to ViT-B/16 (Dosovitskiy et al., 2021). We fix the Adam optimizer and its hyperparameters to values suggested in (Ilharco et al., 2021).

We primarily ablate over only two things: maximum learning rate with cosine learning schedule and warm up iterations for sequential training. For choosing other hyperparameters, we follow the OpenCLIP library (Ilharco et al., 2021).

## D.3 Replay sizes with Exp and Equal strategies

We default to using 2D size of data where D represents incoming data size from new time step. As described in the main text, for -Exp, we reduce the buffer size by half of what we used at old time step and use rest of the half as data from previous time step. App. C.3 lists the dataset sizes for each benchmark which dictate the exact buffer sizes.

## E  Results with Other Continual Learning Methods

### E.1  Results with EWC Method

As proposed in the original work Kirkpatrick et al. (2017), we implement EWC method where we optimize the following loss:

$$\mathcal{L}_{EWC}(\theta) = \mathcal{L}(\theta) + \sum_i \frac{\lambda_{EWC}}{2} F_i(\theta_i - \theta_{t-1,i})^2 \,,$$

where $\mathcal{L}(\theta)$ is the standard contrastive loss on data from time step $t$, $F_i$ is the $i$-th diagonal entry of the fisher information matrix, and $\theta_{t-1}$ are the frozen parameters from previous time step. We perform experiments with different values of $\lambda_{EWC} \in \{1, 10, 100, 400\}$ (see Table 18).

Table 18: **Zero shot performance on our time-continual benchmarks with EWC.** * and ** denote methods that violate the compute budget. For static tasks, we tabulate accuracy of the models obtained on the final timestamp. For dynamic tasks, we tabulate forward/backward transfer and ID performance on retrieval tasks (Sec. 2.3). We observe that EWC performs worse than Sequential, Patching and LwF.

| Benchmark | Method | Compute (MACs) | Static Tasks | | | | Dynamic Retrieval Tasks | | |
| --- | --- | --- | --- | --- | --- | --- | --- | --- | --- |
| | | | ImageNet | ImageNet dist. shift | Flickr30k | Average over 28 datasets | Backward Transfer | ID Performance | Forward Transfer |
| TıC-DataComp (M) | Sequential | $3.0 \times 10^{18}$ | 19.2 | 16.4 | 16.4 | 15.0 | 25.7 | 26.4 | 14.9 |
| | Patching | $3.0 \times 10^{18}$ | 19.3 | 16.8 | 18.5 | 14.7 | 26.9 | 25.4 | 14.5 |
| | LwF* | $3.8 \times 10^{18}$ | 19.2 | 16.5 | 17.7 | 14.3 | 25.6 | 26.6 | 14.9 |
| | EWC ($\lambda_{EWC} = 1$)* | $3.6 \times 10^{18}$ | 18.7 | 16.3 | 16.2 | 15.1 | 25.5 | 26.4 | 14.8 |
| | EWC ($\lambda_{EWC} = 10$)* | $3.6 \times 10^{18}$ | 18.1 | 15.8 | 16.8 | 14.7 | 24.8 | 25.7 | 14.4 |
| | EWC ($\lambda_{EWC} = 100$)* | $3.6 \times 10^{18}$ | 17.6 | 15.4 | 16.3 | 14.8 | 24.4 | 25.4 | 14.3 |
| | EWC ($\lambda_{EWC} = 400$)* | $3.6 \times 10^{18}$ | 17.0 | 15.0 | 16.4 | 14.3 | 24.1 | 24.9 | 14.0 |

### E.2  Results with Oversampling + Counting Based Sampling Method

In this section, we perform ablation on Cumulative-Equal. In particular, we made the following two modifications: (i) *Count based sampling*: Instead of random sampling, we implemented the count-based subsampling that prioritizes not/less used examples; (ii) *Oversampling*: We oversampled data from old timesteps with ratio inversely proportional to the ratio of examples, i.e., if the old data is of size D/2 and the new data is of size D, then we upsample old data with 2:1 ratio.

However, we observe that this method doesn't improve performance over Cumulative-Equal and in fact hurts the performance slightly (see Table 19). We hypothesize that this can be due to a decreasing

marginal utility of labeled data as highlighted in Cui et al. (2019). Their work argues that due to information overlap among data, as the number of samples increases, the marginal benefit a model can extract from the data diminishes. As a result, Cui et al. (2019) proposed using of "effective sample size" instead of the actual number of samples to obtain the ratio used to perform re-sampling or re-weighting. In particular, the expression of "effective sample size" is given by $E_n = \frac{1-\beta^n}{1-\beta}$ where $n$ is the original sample size and $\beta$ is a hyperparameter that Cui et al. (2019) selects from $\beta \in \{0.9, 0.99, 0.999, 0.9999\}$.

For different time steps, we leverage this expression of $E_n$ to calculate the effective number of samples. In our settings (even at small scales), our datasets contain an order of 100k image-text pairs even after subsampling data from old time step. For example, with -Equal baseline, when training on the last time step (i.e., 2022), the smallest dataset (i.e., 2016) is of approximately 400k samples. Plugging in the expression for effective sample size from Cui et al. (2019), we observe that for all $\beta \in (0, 0.99999)$, the ratio of effective sample sizes for different time steps remains close to 1. This may highlight why our naive over-sampling strategy doesn't improve over no-oversampling.

Table 19: **Zero shot performance on our time-continual benchmarks with oversampling and counting-based sampling.** * and ** denote methods that violate the compute budget. For static tasks, we tabulate accuracy of the models obtained on the final timestamp. For dynamic tasks, we tabulate forward/backward transfer and ID performance on retrieval tasks (Sec. 2.3).

| Benchmark | Method | Compute (MACs) | Static Tasks | | | | Dynamic Retrieval Tasks | | |
|---|---|---|---|---|---|---|---|---|---|
| | | | ImageNet | ImageNet dist. shift | Flickr30k | Average over 28 datasets | Backward Transfer | ID Performance | Forward Transfer |
| **TiC-DataComp** (M) | Sequential | $3.0 \times 10^{18}$ | 19.2 | 16.4 | 16.4 | 15.0 | 25.7 | 26.4 | 14.9 |
| | Cumulative-Equal (Counts + OS) | $3.0 \times 10^{18}$ | 18.1 | 15.3 | 14.3 | 16.5 | 28.9 | 23.7 | 14.2 |
| | Cumulative-Equal | $3.0 \times 10^{18}$ | 22.1 | 18.4 | 19.2 | 17.1 | 31.8 | 26.8 | 15.1 |

# F    RESULTS WITH NEW EVALUATION METRICS ON DYNAMIC TASKS

Recall, $T$ represent the number of time steps for which we have data. For each training method, we generate a total of $T$ models, each corresponding to the end of training at a particular time step. For each model and a dynamic evaluation task, we obtain $T$ performance values. We represent these values using the performance matrix $\mathcal{E}$, where each entry $\mathcal{E}_{i,j}$ signifies the performance of the model obtained after observing training data at time step $i$ when evaluated on a dataset from time step $j$. Defining backward metrics as in Sec. 2.2 involves averaging the entries in the upper and lower diagonal of our performance matrix $\mathcal{E}$, i.e., it was calculated as the average of time steps before each training step (i.e., the lower triangular of $\mathcal{E}$), i.e., $\frac{\sum_{i \geqslant j} \mathcal{E}_{ij}}{(T(T-1))/2}$. This backward transfer metric has been used in prior works Lin et al. (2021). However, this approach inadvertently resulted in the backward transfer metric being influenced by later evaluation time steps resulting in backward transfer performance numbers slightly larger than ID performance.

To address this issue, we've revised our metric calculation method to metric as in Díaz-Rodríguez et al. (2018). Now, we normalize the data in each row, which corresponds to evaluation time steps by subtracting the ID performance. This adjustment ensures a more balanced and accurate representation across all training time steps. In particular, our updated forward and backward transfer metrics can be summarized as:

- *Backward transfer*: Let $\mathcal{B}_i$ denote the average performance on evaluation tasks before time $i$, then we define backward transfer as average of $\mathcal{B}_i$ across each training step, i.e., $\sum_{i=2}^{T} \frac{\sum_{i \geqslant j} \mathcal{E}_{ij} - \mathcal{E}_{ii}}{T(T-1)/2}$

- *Forward transfer*: Let $\mathcal{F}_i$ denote the average performance on evaluation tasks after time $i$, then we define forward transfer as average of $\mathcal{F}_i$ across each training step, i.e., $\sum_{i=1}^{T-1} \frac{\sum_{i \leqslant j} \mathcal{E}_{ij} - \mathcal{E}_{ii}}{T(T-1)/2}$

Table 20: **Zero shot performance on our time-continual benchmarks.** * and ** denote methods that violate the compute budget. For dynamic tasks, we tabulate forward/backward transfer and ID performance on retrieval tasks with updated metrics as defined in App. F.

| Benchmark | Method | Compute (MACs) | Dynamic Retrieval Tasks | | | | |
|---|---|---|---|---|---|---|---|
| | | | Backward Transfer | ID Performance | Forward Transfer | Relative Backward Transfer | Relative Forward Transfer |
| **T1C-YFCC** | Restart | $3.4 \times 10^{18}$ | 13.2 | 41.4 | 18.6 | −29.8 | −21.2 |
| | Sequential | $3.4 \times 10^{18}$ | 42.2 | 48.4 | 23.7 | −9.5 | −21.5 |
| | Patching | $3.4 \times 10^{18}$ | 44.7 | 53.4 | 24.5 | −15.6 | −22.0 |
| | Cumulative-Exp | $3.4 \times 10^{18}$ | 60.4 | 60.1 | 27.1 | −9.8 | −23.0 |
| | Cumulative-Equal | $3.4 \times 10^{18}$ | 60.4 | 60.4 | 27.1 | −10.3 | −23.0 |
| | Cumulative-All | $3.4 \times 10^{18}$ | **66.4** | 60.2 | **27.6** | −4.1 | −22.4 |
| | LwF* | $4.1 \times 10^{18}$ | 36.6 | 56.0 | 23.2 | −27.4 | −24.9 |
| | Cumulative-All* | $3.6 \times 10^{18}$ | **66.8** | **60.3** | **27.6** | −3.9 | −22.4 |
| | Oracle** | $8.5 \times 10^{18}$ | 66.1 | 61.8 | 26.9 | −6.6 | −24.0 |
| **T1C-RedCaps** | Restart | $3.4 \times 10^{18}$ | 21.3 | 25.4 | 22.4 | −4.5 | −2.7 |
| | Sequential | $3.4 \times 10^{18}$ | 33.0 | 33.6 | 27.5 | −3.8 | −3.0 |
| | Patching | $3.4 \times 10^{18}$ | 34.8 | 34.8 | 27.8 | −3.9 | −3.0 |
| | Cumulative-Exp | $3.4 \times 10^{18}$ | 44.5 | 42.0 | 32.6 | −3.0 | −4.0 |
| | Cumulative-Equal | $3.4 \times 10^{18}$ | 44.4 | 42.0 | 32.6 | −3.0 | −4.0 |
| | Cumulative-All | $3.4 \times 10^{18}$ | **48.9** | **43.2** | **33.4** | −0.6 | −3.5 |
| | LwF* | $4.1 \times 10^{18}$ | 35.4 | 36.0 | 28.4 | −4.6 | −3.7 |
| | Cumulative-All* | $3.6 \times 10^{18}$ | **49.0** | **43.4** | **33.4** | −1.0 | −3.5 |
| | Oracle** | $8.5 \times 10^{18}$ | 48.5 | 43.1 | 33.4 | −1.0 | −3.4 |
| **T1C-DataComp (M)** | Sequential | $3.0 \times 10^{18}$ | 25.7 | 26.4 | 14.9 | −4.7 | −7.6 |
| | Patching | $3.0 \times 10^{18}$ | 26.9 | 25.4 | 14.5 | −1.9 | −7.4 |
| | Cumulative-Exp | $3.0 \times 10^{18}$ | 31.7 | 27.1 | **15.2** | 0.3 | −7.6 |
| | Cumulative-Equal | $3.0 \times 10^{18}$ | 31.8 | 26.8 | 15.1 | 0.9 | −7.6 |
| | Cumulative-All | $3.0 \times 10^{18}$ | 33.8 | 26.4 | 15.1 | 3.5 | −7.3 |
| | LwF* | $3.8 \times 10^{18}$ | 25.6 | 26.6 | 14.9 | −4.8 | −8.0 |
| | Cumulative-All* | $3.9 \times 10^{18}$ | **36.7** | **28.3** | **15.5** | 3.0 | −7.3 |
| | Oracle** | $1.2 \times 10^{19}$ | 34.9 | 27.8 | **15.6** | 2.5 | −7.7 |
| **T1C-DataComp (L)** | Sequential | $2.7 \times 10^{19}$ | 52.6 | **58.4** | 41.1 | −8.7 | −14.4 |
| | Patching | $2.7 \times 10^{19}$ | 55.2 | 57.5 | 40.9 | −4.9 | −13.9 |
| | Cumulative-Exp | $2.7 \times 10^{19}$ | 60.4 | **58.4** | **41.4** | −1.1 | −13.8 |
| | Cumulative-Equal | $2.7 \times 10^{19}$ | 60.9 | 58.2 | **41.4** | −0.3 | −13.8 |
| | Cumulative-All | $2.7 \times 10^{19}$ | 62.1 | 57.3 | 41.2 | 2.2 | −13.5 |
| | Cumulative-All* | $4.1 \times 10^{19}$ | 63.0 | 57.8 | 41.2 | 2.1 | −13.5 |
| | Oracle** | $1.1 \times 10^{20}$ | **64.3** | **58.6** | 41.8 | 2.2 | −13.3 |
| **T1C-DataComp (XL)** | Sequential | $2.7 \times 10^{20}$ | 63.1 | 68.9 | 56.8 | −5.6 | −12.3 |
| | Cumulative-All | $2.7 \times 10^{20}$ | 70.7 | 68.5 | 57.1 | 2.5 | −11.7 |
| | Cumulative-All* | $3.5 \times 10^{20}$ | **71.0** | **68.6** | **57.1** | 2.5 | −11.7 |

