# OpenReview forum: "TiC-CLIP: Continual Training of CLIP Models"
_ICLR.cc/2024/Conference — ICLR 2024 poster_

### Official Review · Reviewer_R93w · 2023-10-27

**Soundness:** 3 good
**Presentation:** 3 good
**Contribution:** 3 good
**Rating:** 6
**Confidence:** 5

**Summary:**

This paper presents a study on training CLIP with time-evolving data in an incremental manner. The authors timestamp the training data spanning 2014-2022, treating each timestamp as a distinct incremental learning step. Their analysis delves into the backward and forward compatibility of CLIP as it undergoes training, leading to some findings.

Firstly, the research demonstrates that employing simple small replay techniques effectively mitigate forgetfulness, an insight in the context of continual learning. Secondly, the study reveals an unsurprising yet noteworthy performance gap: CLIP underperforms on future, unseen data.

------------------------- After Rebuttal ----------------------------------------------------------------------------------------------

Dear authors,

I did not imply linear probing, but more downstream task adaptation, since CLIP has been applied to almost all computer vision tasks by now.

I went through all the reviews and the rebuttal. I was optimistic about this paper prior to rebuttal. My optimism remains.

**Strengths:**

S1:
This paper addresses a critical and, to the best of my knowledge, an open issue in continual foundation model training. The challenge lies in the impracticality of re-training large foundation models like ChatGPT or CLIP, highlighting the necessity for continual learning solutions. Despite this urgency, there's a notable absence of organized datasets for the community to tackle this problem. While the CLEAR benchmark exists, it has not gained traction and lacks essential components like Web captions. This benchmark, being a captioned version of CLEAR that evolves over time, fills a gap, offering a valuable resource for researchers in the continual learning domain.

S2:
The authors conduct insightful analyses using fundamental regularization and replay techniques. Unsurprisingly, the results show that preserving and replaying a portion of the data during incremental learning effectively mitigates forgetfulness. This empirical validation underscores the importance of such techniques in preserving model performance over time, providing a practical and valuable contribution to the field.

S3:
The paper's central discussion on the fine-tuning cost in terms of compute MAC carries some value. This perspective equips researchers with a framework to explore low MAC solutions, emphasizing a nuanced approach rather than blindly minimizing the forget ratio.

**Weaknesses:**

W1: No method, only benchmarking

The paper is commendable in its focus as a benchmark study, emphasizing data and existing baselines without introducing new methodologies. The utilization of well-known and straightforward baselines is executed competently. However, the paper could have significantly bolstered its strength by incorporating benchmarks from related works such as Continual-CLIP [1] and the methods outlined in "Robust fine-tuning of zero-shot models" [2]. Including these comparisons would have provided a more comprehensive evaluation, highlighting the benchmark's effectiveness in contrast to existing state-of-the-art approaches.

W2: No fine-tuning, only pre-training

An additional area of improvement lies in the paper's scope, particularly concerning the fine-tuning stage. While the study adeptly focuses on the continual pre-training phase, it would have been more insightful and impactful to extend the analysis to the fine-tuning stage. Specifically, investigating how well the obtained checkpoint at a given time transfers to standard vision benchmarks, beyond simple retrieval tasks, would have provided a more nuanced understanding of the model's performance. This consideration is essential for gauging the real-world applicability and adaptability of the continual learning approach outlined in the paper.


[1] Preventing Zero-Shot Transfer Degradation in Continual Learning of Vision-Language Models

[2] Robust fine-tuning of zero-shot models

**Questions:**

No questions so far.

---

> ### Author Response · Authors · 2023-11-14
> **Response to Reviewer R93w**
>
> We thank the reviewer for their positive feedback on our work. We are glad to see the reviewer appreciating the importance of our continual learning benchmark and our experimental protocol.
>
> > **No method, only benchmarking. The paper is commendable in its focus as a benchmark study … However,... incorporating …  related works such as Continual-CLIP [1] and  … "Robust fine-tuning of zero-shot models" [2]. Including these comparisons … highlighting …  effectiveness in contrast state-of-the-art approaches.**
>
> We thank the reviewer for their appreciation of our benchmark study. We would like to clarify that we indeed perform experiments with methods close to both the references the reviewer pointed out. In particular, we perform experiments with Patching (Ilharco et al., 2022) and LwF (Li & Hoiem, 2017; Ding et al., 2022). To clarify: (i) We employ Patching which is a follow up of Wise-FT that adapts Wise-FT to continual learning settings where we observe more than 1 task in sequence; (ii)  We adapt LwF for multimodal models which closely resembles the method in Zheng et al. 2023 (equation 4 performs LwF on both text and vision backbone).
>
> > **No fine-tuning, only pre-training.  An additional area of improvement lies in the paper's scope, particularly concerning the fine-tuning stage … Specifically, investigating how … checkpoint at a given time transfers to standard vision benchmarks, beyond simple retrieval tasks … This consideration is essential for gauging the real-world applicability and adaptability … .**
>
> We perform zero-shot evaluations with 28 existing standard vision benchmarks to evaluate our CLIP models (results are in Table 2 and the standard 28 datasets are listed in Table 4). Our observations highlight that continual pretraining improves on standard tasks (see Figure 1-right and Table 2). Moreover, existing work highlights that zero-shot evaluation is strongly correlated with linear-probe performance (Gadre et al. 2023; Figures 16,17 in Section O). This finding, together with our zero shot evaluation results on 28 datasets, highlight the real-world applicability of continual pretraining for linear probing evaluations.
>
> If we misunderstood your comment, we request the reviewer to provide clarification. We would be happy to run additional experiments if the reviewer has suggestions. We are also happy to run linear probing experiments if the reviewer suggests that these comparisons would strengthen the paper.
>
> We would also be happy to answer any further questions you have. If you do not have any further questions, we hope that you might consider raising your score.

---

> > ### Author Response · Authors · 2023-11-23
> > **Follow up**
> >
> > We wanted to follow up and gently ask if we are able to address your concerns in our response. As the discussion period is nearing an end, we would appreciate any updates or further questions you may have, and if not, we hope you might consider raising your score. We thank you for your time in advance!

---

### Official Review · Reviewer_RTur · 2023-10-31

**Soundness:** 3 good
**Presentation:** 3 good
**Contribution:** 2 fair
**Rating:** 5
**Confidence:** 3

**Summary:**

This paper proposes a benchmark for continual learning using large pretrained vision-language models, e.g., CLIP. The authors construct multiple datasets with timestamp information to evaluate the performance of existing methods on time-continuous data. The experimental results show that the rehearsal-based approach can reduce computation while achieving similar performance as the oracle algorithms.

**Strengths:**

1. The authors construct multiple datasets with time information based on existing datasets for continual learning settings. This is an non-trivial contribution for continue learning to evaluate the effectiveness of algorithms when facing natural distribution shifts.
2. This paper is well-written and easy-to-follow.

**Weaknesses:**

1. This benchmark lacks various types of continual learning methods [1]: elastic weights consolidation methods, progressive neural network methods, dynamic architecture methods, etc. Therefore, the experiments of this benchmark is relative weak and insufficient.
2. This paper lacks some in-depth analysis of vision-language models solving continual learning. Vision-language models enable various novel model tuning paradigms, such as prompt tuning, vision prompt tuning, parameter-efficient tuning, etc. If these aspects are ignored when discussing the solving of continual learning with the CLIP model, then this benchmark is a bit over-claimed.

**References:**

[1] Da-Wei Zhou, Fu-Yun Wang, Han-Jia Ye, De-Chuan Zhan: PyCIL: a Python toolbox for classincremental learning. Sci. China Inf. Sci. 66(9) (2023)

**Questions:**

Please refer to questions in weakness.

---

> ### Author Response · Authors · 2023-11-14
> **Response to Reviewer RTur**
>
> We thank the reviewer for their feedback. We are glad to see that the reviewer finds our continual learning benchmarks as non-trivial contribution.
>
> > **This benchmark lacks various types of continual learning methods [1]: elastic weights consolidation methods, progressive neural network methods, dynamic architecture methods, etc. Therefore, the experiments…is relative weak and insufficient.**
>
> While we included references to these papers, we do not compare with other methods such as  elastic weights consolidation methods and  progressive neural network methods because these methods significantly increase the computation cost (refer to Appendix A.1). Several existing works argue that in constrained-compute scenarios, these methods perform poorly than other CL methods, e.g., LwF and Patching, that are computationally more efficient (Ilharco et al. 2022, Wallingford et al. 2023). We did compare with LwF which often performs empirically better than or similar to EWC (Zheng et al. 2023, Ni et al. 2023). Our findings highlight that this method increases the compute and doesn't improve over our baselines.
>
> Moreover, in our work, we show that simply by increasing the compute budget by 1.2$\times$ for cumulative baseline, we can bridge the gap completely and achieve performance close to or better than the Oracle method. In contrast, methods like EWC significantly increase the compute budget, i.e., at least by a factor of 2 (since we need to compute gradients for two models) excluding the time to estimate the eigenvalues which can also incur significant costs (Ding et al. 2022).
> We also thank the reviewer for sharing the reference on the toolbox. We have now included this in our work.
>
> > **This paper lacks  … vision-language models solving continual learning. Vision-language models enable various novel model tuning paradigms, such as prompt tuning, parameter-efficient tuning, … aspects are ignored …  then this benchmark is a bit over-claimed**.
>
> We would like to clarify and further highlight the distinct nature of our work in comparison to existing research. Unlike the majority of prior studies which primarily focus on task/domain/class incremental setups, our work diverges significantly. In these previous works, such as those by Ni et al. 2023, Ding et al. 2022, and Zheng et al. 2023, the emphasis has been on handling small-scale downstream task data, typically of a very small order than pretraining dataset size. These methods tend to produce models specialized in individual domains. However, in our work, we focus on pre-training (general knowledge acquisition) of the base model with continually evolving internet data. We create the first benchmark with 12.7 billion timestamped img-text pairs that enables systematic study of continual pretraining of CLIP models. This benchmark paves the way for a systematic exploration of the continual pretraining methods  in CLIP models.
>
> While fine-tuning a limited set of parameters may be an effective strategy for tailoring a model to a confined dataset from a specific domain, this approach is less viable for maintaining and updating "general-purpose" foundational models (e.g., CLIP) with dynamically evolving internet data. In our setup, the proportional increase in new data relative to the pre-existing dataset necessitates comprehensive model updates.
>
> *(New result)* As per your suggestion, we performed an additional experiment to assess the efficacy of updating selected model parameters versus full-scale fine-tuning with sequential training on the TiC-YFCC dataset. In particular, we compare LiT [1] where we perform full-finetuning on the first time step and then perform LiT for subsequent time steps on Tic-Datacomp (M). We observed that while updating a few parameters does avoid catastrophic forgetting on old timestamps, this happens at the cost of reduced performance on all tasks. For instance, the final Imagenet performance drops substantially from 24.0 (w full finetuning) to 7.3 (w LiT).  This highlights limitations of finetuning only a small number of parameters while trying to keep general purpose foundation models up to date.
>
> Moreover, we would like to emphasize that the main goal of the paper is NOT to introduce a new continual learning technique, rather introduce the first benchmark for continual pretraining where we pretrain foundation models as internet data evolves. For this setup, we present extensive results using standard continual learning approaches that are suitable for pre-training and highlight a simple method that enables continual pretrain of CLIP models without retraining models from scratch when new data arrives.
>
> We would be happy to answer any further questions you have. If you do not have any further questions, we hope that you might consider raising your score.
>
> ----
> [1] Zhai et al. 2021. LiT: Zero-Shot Transfer with Locked-image text Tuning
>
> [2] Wallingford et al. 2023. Fluid: A Unified Evaluation Framework for Flexible Sequential Data

---

> > ### Comment · Reviewer_RTur · 2023-11-18
> >
> > I understand your difficulties with comparison methods, but these efforts are necessary as this is a paper releasing a benchmark. I decided to raise my score to show my appreciation for your work but I still can not fully support accepting this paper.

---

> > > ### Author Response · Authors · 2023-11-19
> > > **Reply to Reviewer RTur**
> > >
> > > We thank the reviewer for their reply. We are running additional experiments with EWC and we will update by tomorrow with their results. Please let us know if you have suggestions for any other methods that we should include. We are asking this so that we can implement other suggestions given the approaching deadline for the reviewer-author discussion period.

---

> > > > ### Author Response · Authors · 2023-11-22
> > > > **Response to Reviewer RTur**
> > > >
> > > > We have now performed additional experiments with EWC method. We include results on TiC-Datacomp (medium) scale in Appendix E.1 in Table 17.  We observe that EWC performs worse as compared to Sequential and Patching baselines (which are computationally cheaper).
> > > >
> > > > | Benchmark       | Method                  | Compute (MACs)   | ImageNet | ImageNet dist. shift || Flickr30k | Average over 28 datasets | | Backward Transfer | ID Performance | Forward Transfer |
> > > > |-----------------|-------------------------|------------------|----------|----------------------|--------------|-----------|--------------------------|-------------------------|-------------------|----------------|------------------|
> > > > | **TIC-DataComp (M)** | Sequential              | 3.0 x 10^18      | 19.2     | 16.4                 |              | 16.4      | 15.0                     |                         | 25.7              | 26.4           | 14.9             |
> > > > |                 | Patching                | 3.0 x 10^18      | 19.3     | 16.8                 |              | 18.5      | 14.7                     |                         | 26.9              | 25.4           | 14.5             |
> > > > |                 | LwF*                    | 3.8 x 10^18      | 19.2     | 16.5                 |              | 17.7      | 14.3                     |                         | 25.6              | 26.6           | 14.9             |
> > > > |                 | EWC (λ_EWC = 1)*        | 3.6 x 10^18      | 18.7     | 16.3                 |              | 16.2      | 15.1                     |                         | 25.5              | 26.4           | 14.8             |
> > > > |                 | EWC (λ_EWC = 10)*       | 3.6 x 10^18      | 18.1     | 15.8                 |              | 16.8      | 14.7                     |                         | 24.8              | 25.7           | 14.4             |
> > > > |                 | EWC (λ_EWC = 100)*      | 3.6 x 10^18      | 17.6     | 15.4                 |              | 16.3      | 14.8                     |                         | 24.4              | 25.4           | 14.3             |
> > > > |                 | EWC (λ_EWC = 400)*      | 3.6 x 10^18      | 17.0     | 15.0                 |              | 16.4      | 14.3                     |                         | 24.1              | 24.9           | 14.0             |
> > > >
> > > >
> > > >
> > > > If the reviewer suggests, we can include results with EWC on TiC-YFCC and TiC-Redcaps in the final version.

---

> ### Author Response · Authors · 2023-11-22
> **Checking in**
>
> We thank the reviewer for their engagement in the rebuttal process. Since the discussion period is coming to an end today, we wanted to follow up on our response from yesterday. While we understand that it has been only several hours since our last response, we wanted to ask if you have any major additional questions or concerns left, and if not, we hope you might consider raising your score.

---

### Official Review · Reviewer_fC5h · 2023-11-02

**Soundness:** 4 excellent
**Presentation:** 4 excellent
**Contribution:** 3 good
**Rating:** 8
**Confidence:** 4

**Summary:**

This work introduces new pretraining datasets and dynamic downstream tasks for web-scale continual learning using CLIP on natural distribution shifts.

Datasets introduced by ordering metadata by time:
- **TiC-DataComp**: 12.7 billion image-text pairs from datacomp-xlarge ordered by first seen snapshot (monthly increments available from 2014-2022), with 7 timesteps from (2014-2016) and every subsequent year.
- **TiC-YFCC**: 15M image-text pairs on subset of YFCC100M which have captions and timestamps. Data from 2008–2014, with 4 timesteps involving (2008-2011) and every subsequent year.
- **TiC-RedCaps**: 12M image-text curated from Reddit in 2011-2020 ordered by creation timestamps of posts, with 4 timesteps (2011-2017) and three subsequent years.

Evaluation

**Static Tasks**: Zero-shot evaluation on a set of 28 downstream tasks similar to (Radford et al 2021)

**Dynamic Tasks**: On a small subset of samples reserved for testing at each timestep,
- (i) T2I retrieval for samples in a given timestep
- (ii) Classification on a LAIONNet-like 1000 class dataset (filtered by sentence embedding). Similar in nature to a scaled up CLEAR dataset.

**Metric**: Use the checkpoint after every timestep and perform classification across different timesteps to evaluate In-domain Acc, Backward Transfer and Forward Transfer.

**Primary Findings**:
1) Continual training saves Cumulative* ~3x the cost for TiC-Datacomp compared to training-from-scratch (Oracle**), with a similar savings  shown for ImageNet IID-incremental.

2) Comparison between Sequential and Cumulative-all on TiC-Datacomp: This highlights sequential has
- significant catastrophic forgetting (low backward transfer performance)
- similar forward transfer performance – indicating catastrophic forgetting does not impact generalization to new distributions

This entails:
- poor performance static benchmarks, having poor backward transfer performance leads to a performance hit on static benchmarks which have old data (supported by other evidence: Fig 9&11)
- Consequently adding a memory buffer seems to help static tasks.
3) Patching helps a lot more than traditional continual learning methods like LwF when training without past data, indicating different continual learning approaches might help in these settings.

4) Different models work better for static and dynamic evaluation tasks.

**Strengths:**

S1) **Tackles an important problem** [Critical]: This work correctly highlights the need to shift focus in continual learning and introduces time-evolving benchmarks for evaluating continual pretraining which turns out is quite important. I really liked the dynamic retrieval and the classification task design. Retrieval captures performance shifts in time by new concepts and distribution shifts, whereas classification task ablates the performance gap caused due to new things (e.g. covid) by choosing the same 1000 classes

S2) **Insightful analysis** [Critical]: This work was a delight to read. I liked this work motivating a case where optimizing models for old datasets might lead to continuously worse performance on current-day tasks. Similarly, summary (4) indicates picking continual SSL training strategies based on downstream tasks might lead to poor design choices (best models worse than intended), shown in experiments on best-pool filtering.

S3) **High-quality work** [Critical]: The contributions seem fairly clear, experiments investigate interesting related claims well and experimentation is quite extensive.

**Weaknesses:**

W1) **Sequential and cumulative models behave quite differently between TiC-YFCC15M and TiC-Datacomp** [Critical]
- The paper nicely illustrates that YFCC15M has strong distribution shifts in Figure 15.
- However, does TiC-DataComp have significant distribution shifts?
    - The case for continually training CLIP primarily relies on Datacomp-like data having strong distribution shifts.
    - I suspect the case there is far weaker than YFCC15M (I am worried it's too small to make this setting exciting).
- Can the authors create a plot similar to Figure 15 for TiC-DataComp or some other mechanism to analyze distribution shift in TiC-Datacomp?

W1.1) Further unexplained variations between Sequential and -All:
- In TiC-YFCC15M and TiC-RedCaps, the performance of forward transfer is significantly affected, but that is not the case in TiC-Datacomp.
- Similarly, the gap between backward transfer and ID retrieval is far smaller between them in TiC-Datacomp. Why is this the case?

W1.2): Fig1 (mid) and Figure 10 present a different picture than Table 2.
- Comparing backward transfer retrieval to ID performance for Cumulative exp (oracle) in Table 2
    - Intuitively, ID performance should be higher (equal).
- To what degree is the drop in forward transfer attributable to tasks getting harder vis-a-vis encountering new distributions.

Hypothesis: Past tasks are easier! Can the authors help shed some light on why the difference?

W2) **Major results simply not present** [Critical]

I cannot find any results for LAIONNet-like dynamic classification task. Did I miss something?

Other missing ablations:
- a) “Same maximum LR works best across all runs when using cosine schedule” – Didn't find supporting evidence.
- b) “Warm up helps training on data from the first time step, but hurts on subsequent time steps.” – After correcting the minor shift issue, the table seems to robustly support the opposite conclusion. This seems surprising to me! I would have not expected warmup to help as the init is a very good one. Warmup mostly mitigates over-updating to high gradients when starting from a poor (random) init in my view.
- c) “Given a fixed buffer size for each past step, we observe minimal to no difference between random subsampling and other strategies.” – Didn't find supporting evidence.

W3) **Serious Design Flaws in Continual Learning Setup** [Critical]

W3.1) *4-7 timesteps are too few*
- I am worried the major findings would significantly change when tested on 20/50 timesteps but with the same computational cost per timestep, as the streams become far more non-i.i.d in that case.
- Why? Because of memory buffers become far smaller, that it becomes very hard in my experience to bridge the gap between CL methods and -all.
- Similarly, I am concerned there will be a far larger gap between -all and Oracle.

This critically affects some of the core findings presented in this work.

W3.2) *Is memory constrained?*
- For TiC-RedCaps and TiC-YFCC15M, all and exp/equal only differ at last timestep (Until timestep 3, they all can store all the data). This grossly undermines the comparisons as even at the last timestep, they still store 2/3rd of the data. It is very surprising that the difference in training for timestep causes such a big gap! If this is caused by D/3 data missing at the last timestep, the findings given 20-50 will probably be quite different!
- This issue is mitigated in TiC-DataComp as at last timestep, equal would store D/3 samples per timestep (2D/6D) which is indeed much  smaller! (Minor note: Am I correct that buffer conserved here is 2.3x smaller, not the claimed 3.5x smaller? -- The replay+current is 3D/7D).

W3.3) *Replay buffers not utilized effectively, computation implicitly constrains memory.*
- I worry most of the decline in performance is due to random sampling, and hence wanted to see supporting evidence for W2.c). I make my case below:
- *Why?* This seems to be a case of imbalance data across timesteps due to the buffer constraints.
   - If trained without oversampling less-represented data, I am not surprised to see a bias against replay data (i.e. poor backward transfer).
- *Why not data loss constraining replayable sample?* I focus on TiC-Datacomp because there is a significant gap in replay buffers here
   - In TiC-Datacomp-medium, 35k iterations with 4096 samples barely allows ~1 pass through the 128M samples.
   - Hence, there seems to be a far stronger implicit memory constraint, as even after storing kD samples at timestep k, the network can only pass through 1/k fraction of samples from each past timestep at timestep k (training with oversampling).
   - Given a fixed buffer size, retaining equal samples of past data still stores 2/(k-1) samples which is far higher than the limitation introduced by computational constraints. It seems possible to train on mostly unique samples, mimicking -all.

Hence, the gap in performance between -equal and -all should have been minimal (which seems like a serious design issue-- alleviated by far smaller buffers in the 20-50 timestep setup discussed above). However, the fact that gap still exists seems to compound the issue, and I worry it is due to poor sampling.

W4) **Little discussion of past work despite similar findings/parallels** [Important]

Although I fully admit the dominant setups in CL are quite synthetic and need improvement, as correctly pointed out by the authors.
- (Cai et al., 2021) also has analysis covering several aspects of interest here, e.g. YFCC distribution shift plot (see their Fig 2), learning rate modulation, comparing impacts of replay buffer, web-scale training and similar compute constraints as here but on 600K timesteps. This avoids promoting CL methods which work only on a short timescale and degrade quickly when scaled beyond 5-20 timesteps (i.e. my worry about the 4-7 timesteps).
- Finding (1) on Imagenet IID seems out-of-place here, and has detailed precisely in several works, e.g. “Computationally Budgeted Continual Learning: What Does Matter?”, CVPR23. Other works like “One Pass ImageNet” NeurIPS-W 2021 have analysis on Sequential/Cumulative variants as discussed here on Imagenet but these papers are not discussed.
- Other works which have interesting investigations with computational limits (Bornschein et al., 2022) and (Jang et al., 2022) have nice analysis quite relevant to this work. E.g. hyperparamter optimization as a cost in the compute budget C (will be required when training in the unknown)

**Questions:**

Q1) What is the size of dynamic retrieval and LAIONNet test sets per chunk?

Minor Comments: (Not considered for score)

C1) Metric
- The backward and forward transfer metric utilized here seems to be from (Lin et. al., 2021).
- (Lopez-Paz & Ranzato, 2017) have a very different metric for backward and forward transfer, while (Díaz-Rodríguez et al., 2018) do not introduce any modification to this metric.

Overall, I strongly agree with the motivations and findings from analysis (filtering, CLIP vs OpenCLIP) presented. I worry the continual learning experiments made poor design choices which might seriously impact most of the major findings in this work (W1, W3). If addressed, I will be very happy to increase my score.

Note: I tried to detail why the ask, primarily to minimize superfluous asks which I find annoying from reviewers on my submissions (Did not mean to sound pretentious).

[Edit]: Increased score from 3 to 8 and soundness from 1 to 4, as additional analysis resolve my pointed concerns.

---

> ### Author Response · Authors · 2023-11-14
> **Response to Reviewer fC5h (Part 1)**
>
> We thank the reviewer for their extremely detailed constructive feedback on our work.  It is encouraging to note that the reviewer recognizes the high quality of our paper, its relevance in addressing an important problem, and the depth of our analysis.
>
> We provide detailed answers to your concerns below and we believe that by addressing these concerns we have significantly improved the manuscript.
>
>
> > **W1) Sequential and cumulative models behave quite differently between TiC-YFCC15M and TiC-Datacomp**
>
> We give a brief answer here and provide a details as we answer the sequence of questions below. The primary reasons are two (see Sec B.7 for detailed discussion):
>
> (i) the nature of the distribution shift observed in YFCC and its constructed dynamic retrieval task; we observe that models trained on Tic-YFCC suffer from relatively larger drops due to catastrophic forgetting than Tic-Datacomp (see updated plots for Sequential in Fig 6 in App B.1).
>
> (ii) compute used at each time step per data available at each time step. Overall YFCC is 2x smaller than Tic-Datacomp (M) but the compute we used in both TiC-YFCC and TiC-Datacomp setup is of similar order (in fact, it is slightly higher in TiC-YFCC). We re-ran the experiments for Tic-YFCC by reducing the compute. In the updated runs, we observe that the gap between ID performances of Sequential and Cumulative-All vanishes .
>
> We also emphasize that the goal of our study is to highlight the problem with the distribution shift for data collected from different data sources (e.g., Flickr, Common Crawl, Reddit). While we agree that the distribution shift severity is different in different data sources, we show that the problem exists for all different data sources. Similar to Figure 4 (right), we have now added heatmaps with other methods and datasets on dynamic retrieval evaluation tasks (Figure 6). For a fixed training time (a specific row), we clearly see performance drop on evaluation time steps after the training time step, highlight distribution shift in these benchmarks.
>
> - > **… does TiC-DataComp have significant distribution shift**
>
> Yes, we observed that TiC-Datacomp has distribution shift problems as shown by the performance difference between forward transfer and backward transfer (in Table 2, Figure 4 (right) and Figure 6 (c,d)). We clearly observe that performance on dynamic tasks from future time steps is worse than current or previous time steps. Figure 3 presents some examples highlighting how distribution evolves over time in TiC-Datacomp. For example, with time images from novel concepts appear (e.g., COVID-19).
>
> - > **Can the authors create a plot similar to Figure 15 for TiC-DataComp or some other mechanism to analyze distribution shift in TiC-Datacomp?**
>
> We have included a plot highlighting the nature of the shift in TiC-Datacomp in updated Figure 16. We observe similar trends for FID score on feature distribution for TIc-Datacomp. For TV distance, the distance increases initially and then saturates for later timesteps.
>
> - > **In TiC-YFCC15M and TiC-RedCaps, …  forward transfer is significantly affected, but … not … in TiC-Datacomp. Similarly, gap between backward transfer and ID retrieval is far smaller between them in TiC-Datacomp. Why is this the case?**
>
> We primarily attribute this to the data source differences between these constructed benchmarks. While we agree that the distribution shift severity is higher in TiC-YFCC and TiC-Redcaps, we show that this problem exists even for TiC-Datacomp. We have now included Figure 6 where we plot the performance evaluation matrix $\mathcal{E}$ for all of these datasets.
>
> - > **Fig1 (mid) and Figure 10 present a different picture than Table 2. … In Table 2, intuitively, ID performance should be higher (equal)**
>
> Your observation about the discrepancy in the forward and backward transfer metrics is insightful. Thank you for highlighting this. Originally, our methodology involved averaging the entries in the upper and lower diagonal of our performance matrix $\mathcal{E}. This approach inadvertently results in the backward transfer metric being influenced by later evaluation time steps resulting in backward transfer performance numbers slightly larger than ID performance.
>
> To address this issue, we've revised our metric calculation method by deviating from prior work Lin et al 2021. Now, we normalize the data in each row by subtracting the ID performance as in Díaz-Rodríguez et al., 2018) This adjustment ensures a more balanced and accurate representation across all training time steps. We have also adjusted the forward transfer metric similarly (Table in Appendix ). We have updated the draft with these change (App E). In the final version, we will include these results in addition to the existing results.
>
> We have also included additional plots that capture the performance matrix \mathcal{E} as in Figure 4 (right) on all datasets in the appendix (Figure 6 in App B.1).

---

> ### Author Response · Authors · 2023-11-14
> **Response to Reviewer fC5h (Part 2)**
>
> - > **To what degree is the drop in forward transfer attributable to tasks getting harder vis-a-vis encountering new distributions.**
>
> This is a great question. To elucidate the differences, we compared performance of the Oracle method trained on data from all time steps on evaluation sets from different time steps, i.e., The last row in dynamic evaluation of Oracle method in Figure 4 (right) and in Figure 6 (App B).
>
> Since the models evaluated in the last row of Oracle method are trained on all training data, differences in performance of the Oracle model at the last training time step primarily stem from hardness of distributions at every time step.
>
> We make the following observations. For TiC-YFCC and TiC-Redcaps, we observe an increasing trend in dynamic retrieval performance highlighting that the task becomes easier for future time steps. For TiC-Datacomp, the hardness of the task remains almost the same (slight increase) as the retrieval performance remains almost the same. These observations hint that while we encounter different distributions for all datasets, for TiC-Datacomp the hardness of the tasks in future timesteps remains almost the same, whereas, for TiC-YFCC and TiC-Redcaps, the tasks become easier as time progresses.  This observation hints that for TiC-YFCC and TiC-Redcaps, the quality of images and their corresponding captions improve over time. For example, the average quality of cameras on mobile phones have improved.
>
> > **W2) Major results simply not present**
>
> We have now included all the missing tables in the appendix or references to already existing tables in the appendices.
>
> - >**Same maximum LR works best across all runs when using cosine schedule**
>
> Table 7 in the updated draft provides evidence to this finding (we have added the missing reference in the main paper). Earlier draft also included this table with a typo in one column – we have fixed that.
>
> - >**Warm up helps training on data from the first time step, but hurts on subsequent time steps – After correcting the minor shift issue, the table  … support the opposite conclusion.**
>
> Sorry for the confusion here. We copied the column names incorrectly (you can check the mismatch between column names and the corresponding results between Table 6 and Table 2). We have now fixed this table in the updated draft which matches the stated finding in our paper.
>
> - > **Given a fixed buffer size for each past step, we observe minimal to no difference between random subsampling and other strategies. – Didn't find supporting evidence**
>
> We only performed preliminary analysis for this. In our preliminary experiments, we explored the efficacy of subsampling old data based on the alignment between text and image content from previous time steps. Specifically, when training a model at time step t+1 , we used the model from the end of time step t to assess this alignment. We employed two distinct subsampling methods:
> 1. Retaining half of the data with the lowest alignment scores, based on the premise that these data points might be more challenging to learn and require additional gradient steps.
> 2. Retaining half of the data with the highest alignment scores, under the assumption that these represent higher quality data, as indicated by the stronger alignment between text and image pairs.
>
> We applied these methods to the TiC-YFCC dataset and evaluated their performance against a baseline of random sampling. The outcomes revealed minimal differences: less than 0.2% variation in Imagenet performance and under 0.5% in dynamic retrieval performance across different time steps. Given that these minor improvements came with a significant computational cost—requiring a full forward pass to compute alignment post each training epoch—they exceeded our compute budget constraints. As a result, we opted for random sampling in our research. These findings have been included in the appendix for further reference (App B.4). We leave investigation on improved subsampling techniques for future work.
>
> - > **I cannot find any results for LAIONNet-like dynamic classification task.**
>
> We apologize for missing the table for this result. We have included this in the updated draft and have added a reference to that in the main paper.  The trends described in Sec 3.4 hold true. We have added Table 8 in App B.6 and Figure 15 in C.6.

---

> > ### Comment · Reviewer_fC5h · 2023-11-19
> > **Reply to Rebuttal [Part2]**
> >
> > > Tasks getting harder over time
> >
> > Figure 6 (App B) of Oracle comparing the lower diagonal values can indeed precisely tell this apart. Based on this experiment, authors conclude-
> >
> > For TiC-YFCC and TiC-Redcaps, the task becomes easier for future time steps. For TiC-Datacomp, the hardness of the task remains almost the same (slight increase). Sounds convincing, surprising and thorough. I can't find any holes here!
> >
> > W2) Apart from the results for W2 c) I found the rest of the results, they seemed satisfactory. I shall highlight critical issues in the experiment on W2 c) in the next part.
> >
> > On LAION-Net results:
> > - The gaps in dynamic LAION-Net task seem much smaller than the corresponding ones from the broader retrieval task.
> > - Is it a correct to say that a large part of the gap comes from new concepts rather than a distribution shift on existing concepts? (Assumes the retrieval task was the same, the only difference being conditioning on fixing concepts or not. Would appreciate some sort of discussion of this in the work)
> >
> > Minor note: Would like to have the bestpool results "For TIC-DataComp (XL), we include results with Bestpool filtering." together in this table or alternatively with and without bestpool filtering in Table 5. It was painful to repeatedly scrolling back and forth to compare numbers.

---

> ### Author Response · Authors · 2023-11-14
> **Response to Reviewer fC5h (Part 3)**
>
> > **W3) Serious Design Flaws in Continual Learning Setup**
> - > **4-7 timesteps are too few … major findings would significantly change when tested on 20/50 timesteps**
>
> We acknowledge the reviewer's perspective that extending our analysis to 20/50 timesteps could yield different findings. However, we contend that our chosen 4-7 timestep range (one model a year for up to 7 years) is reflective of certain real-world scenarios, particularly in fields where data is subject to rigorous quality filterings and models are subject to rigorous evaluations between deployments. Such models are expected to adapt to slow changes in the data distribution (year to year) and tolerate faster but small changes (month to month). Examples of such settings include image classification for autonomous driving, where updates are thoroughly tested in controlled environments to ensure safety and compliance with regulations. Even models, like ChatGPT undergo updates at a relatively smaller frequencies (After its initial release about an year ago, GPT-4 was only recently updated with new knowledge).
>
> Moreover, **our work is  the first benchmark for continual training at the scale of 12.7 B image-text pairs** (e.g., CLOC [Cai et al. 2021] has only 39M samples, see others in Table 4). This initial exploration into the realm of 4-7 timesteps  multi-year web-scale data can pave the way for future work to explore the more challenging settings with 50-60 time steps. We have prioritized the large-scale aspect of our benchmark and tapped the largest sources of publicly available image-text data. Our benchmarks effectively span 14 years from 2008-2022. As such, creating a benchmark with more than 14 years with data prior to 2008 might be infeasible as of now as the scale and quality of earlier data is much less.
>
> Alternatively, our benchmark itself provides a testbed to experiment with finer granularity of months (Sec 2.3). Our dataset release will include time stamped data at the granularity of months. We believe that it would be an interesting direction for future work to explore continual learning at finer granularities in our TiC-DataComp benchmark where we have 70-80 time steps. In such regimes, we imagine that one can explore techniques that alternate between full model updates (which can be done less frequently) and finetuning a few parameters to adapt to more rapid distribution shifts (which can be done more frequently).
>
> - >**Is memory constrained? For TiC-RedCaps and TiC-YFCC15M, all and exp/equal only differ at last timestep … This issue is mitigated in TiC-DataComp as at last timestep …  note: Am I correct that buffer conserved here is 2.3x smaller, not the claimed 3.5x smaller? …**
>
> We apologize for confusion here. Table 1 in the draft depicted the correct sizes and there was a minor typo in the text for Cumulative method description in Sec 3. We have updated the draft and reflected the change. In our experiments, we only replay D size of old data, and thus even for TiC-RedCaps and TiC-YFCC15M, we have differences in the last two steps and for TiC-DataComp, we have differences in the last five steps. This also addresses the confusion between 3.5x reduction in memory buffer (3.5x is the correct reduction).
>
> - > **Replay buffers not utilized effectively, computation implicitly constrains memory. I worry most of the decline in performance is due to random sampling, and hence wanted to see supporting evidence for W2.c).**
>
> We have added evidence to the W2c weakness above. We also thank the reviewer for their detailed suggestion on methods to try and for their astute feedback.
>
> The main goal of our work is to construct the continual learning benchmark and evaluate simple but important baselines. We believe that the method suggested by the reviewer is a very interesting direction for future work on how to improve sampling datasets when selecting the replay buffer for continual learning at scale.

---

> ### Author Response · Authors · 2023-11-14
> **Response to Reviewer fC5h (Part 4)**
>
> > **W4) Little discussion of past work despite similar findings/parallels**
>
> We thank the reviewer for providing references to these papers.
>
> - > **(Cai et al., 2021) …l aspects of interest here, learning rate modulation, comparing impacts of replay buffer, web-scale training … but on 600K timesteps. ... CL methods work only on a short scale and degrade beyond 5-20 timesteps.**
>
> Yes, we did reference the work of Cai et al. 2021 in our draft and have now expanded (in App A.2) our discussion on it in the revised draft. We agree that this work provides interesting discussion/analysis for continual learning at a large number of steps. However, our study differs from Cai et al. 2021, in several crucial respects:
>
> (i) Training Methodology: We employ noisy supervision using contrastive loss between image-text pairs, as opposed to the cross-entropy loss used by Cai et al. 2021.
>
> (ii) Scale of Experiments: Our experiments on the TiC-DataComp dataset are orders of magnitude larger, scaling up by 200x.
> These differences introduce unique challenges. The use of contrastive loss (i) necessitates a tailored approach to designing our evaluation studies. The significantly larger scale of our experiments (ii) poses challenges in collecting timestamped data and understanding if and how distribution shifts impact learning at this scale. As mentioned before, we believe that extending the study on our TiC-DataComp benchmark (with 12.7B examples) to a finer granularity as in Cai et al. 2021 is an interesting direction for future work.
>
> - > **Finding (1) on Imagenet IID …  in several works, e.g. “Computationally Budgeted Continual Learning: What Does Matter?” … Other works like “One Pass ImageNet” … have analysis on Sequential/Cumulative variants on Imagenet but  … not discussed.**
>
> We are grateful to the reviewer for suggesting these additional references, which we have now incorporated the suggested papers into our discussion (in App D.1). If the reviewer suggests, we are open to moving the Imagenet results fully to the appendix.
>
> Our Imagenet experiments were primarily inspired by the "loss of plasticity" phenomenon described in Ash and Adams 2020. Their study demonstrates that models sequentially trained on two splits of CIFAR-10 data (initially on 50%, followed by 100% of data) exhibit poorer generalization compared to models trained from scratch on the entire dataset. Since we do not observe this behavior for continual training of CLIP, we investigated the existence of such behaviors on up to 8 splits of Imagenet. Our findings reveal that the simple cumulative baseline remains competitively close to the Oracle model (that benefits from using the full compute budget on the entire pooled training data from the beginning).
>
> Although the referenced works and ours both underscore the effectiveness of continual training on synthetic continual learning setups derived from ImageNet, the specific settings in these studies vary significantly from ours, limiting direct comparisons. A key distinction of our work is the comparison of our final models with an Oracle model, a step not undertaken in the referenced papers as it was not their primary objective.
>
> - >**Other works which have interesting investigations with computational limits (Bornschein et al., 2022) and (Jang et al., 2022) have nice analysis quite relevant to this work. E.g. hyperparamter optimization as a cost in the compute budget C (will be required when training in the unknown)**
>
> Yes, we indeed cite these references in our work and now we have expanded the discussion with these papers. Our methodology maintained consistent base hyperparameters across all timesteps, with the exception of specific ablations on learning rate warm-up and maximum learning rate at a medium scale. Consequently, our experimental framework does not account for the costs associated with hyperparameter selection.
>
> > **What is the size of dynamic retrieval and LAIONNet test sets per chunk?**
>
> We have now added sizes of our evaluation tasks in App C.3.
>
> > **The backward and forward transfer metric utilized … from (Lin et. al., 2021). (Lopez-Paz & Ranzato, 2017) have a … different metric for backward and forward transfer, while (Díaz-Rodríguez et al., 2018) do not … modification to this metric.**
>
> Thanks for highlighting the differences; we have clarified this in the updated draft (in App E). The main commonality in all of our works is the same performance matrix $\mathcal{E}$. While Díaz-Rodríguez et al., 2018 used the same forward transfer metric as ours, the backward transfer metric in their work and metrics in Lopez-Paz & Ranzato, 2017 performed centering by subtracting the ID performance. Note that, as discussed above, we have now updated the metrics to include the metrics used in Díaz-Rodríguez et al., 2018 in our work.
>
>
> We would be happy to answer any further questions you have. If you do not have any further questions, we hope that you might consider raising your score.

---

> ### Comment · Reviewer_fC5h · 2023-11-19
> **Reply to Rebuttal [Part1]**
>
> W1) Sequential and cumulative models behave quite differently between TiC-YFCC15M and TiC-Datacomp
>
> Authors main visualization support is:
> a) We have now included Figure 6 where we plot the performance evaluation matrix for all of these datasets.
>
> with conclusions:
> (i) Tic-YFCC suffer from relatively larger drops due to catastrophic forgetting than Tic-Datacomp
> (ii) On re-running the experiments for Tic-YFCC with reduced compute, the gap between ID performances of Sequential and Cumulative-All vanishes.
>
> The authors additionally provided:
> > (i) We have included a plot highlighting the nature of the shift in TiC-Datacomp in updated Figure 16. We observe similar trends for FID score on feature distribution for TIc-Datacomp. For TV distance, the distance increases initially and then saturates for later timesteps.
>
> On this point, the updated TV-distance plots did not have increasing drift. To ensure that the drift is not simply between the first timestep and the rest, can the authors extend the Datacomp plot the same graph across some years in the middle and the end timestep additionally?
>
> I am very happy/convinced by the rebuttal on this point and tentatively consider this fully resolved. I would be very happy if I could see the Figure 16 plots across 2-3 timesteps before rebuttal deadline to complete this point.

---

> ### Comment · Reviewer_fC5h · 2023-11-19
> **Reply to Rebuttal [Part 3]**
>
> **W3.1) [Too few timesteps]** I am not convinced on this and would push back. Setting this norm would be important for a benchmark paper, as a lot of methods in continual learning fail to mitigate forgetting beyond a few timesteps. Similarly, I am concerned there will be a far larger gap between CL methods and -all, and -all and Oracle. However, with the correction in buffer being D and not 2D this is no longer a critical concern considered for score of the work, just an drawback of this benchmark. If this is specifically acknowledged it would be sufficient for me.
>
>
> **W3.2) [Is memory constrained?]** Resolved fully, thanks!
>
> **[W3.3) Replay buffers not utilized effectively]** This has not been resolved at all, and is a critical drawback. The specific issue pointed out is data-imbalance and using unique samples. The sampling strategy this implies testing with is to balance the number of samples selected between past/current data and prefer deleting the samples which were used for replay over samples which were not when buffer size is constrained. I would strongly recommend testing this correction and presenting results as this is critical for the difference between cumulative-eq and cumulative-all for Datacomp-medium, and in the final version for all datasets.
>
> The authors instead explored unrelated sampling strategy which had little to do with the question and  requires additional forward passes (similar strategies have been described to not work in past literature precisely due to being too expensive).
>
> W3.2 and W3.3) were the primary concerns I had in this work. W3.2 and W1 were addressed, and I shall raise the score to 6 to reflect this. If W3.3 is sufficiently addressed on the smallest Datacomp dataset, I would have no further concerns and can commit to raising my score to 8.

---

> ### Comment · Reviewer_fC5h · 2023-11-19
> **Reply to Rebuttal [Part4]**
>
> All replies except the ImageNet-IID claim seem satisfactory to me (the (Díaz-Rodríguez et al., 2018) part particularly was quite neat, thanks for this!). The ImageNet-IID claims seems to misrepresent OPIN and CB, as I explain below.
>
> > Our findings reveal that the simple cumulative baseline remains competitively close to the Oracle model (that benefits from using the full compute budget on the entire pooled training data from the beginning)... the specific settings in these studies vary significantly from ours, limiting direct comparisons. A key distinction of our work is the comparison of our final models with an Oracle model, a step not undertaken in the referenced papers as it was not their primary objective.
>
> Both works compare with an Oracle model!
>
> [OPIN] As per my understanding, the main comparison in OPIN is between Oracle and continually trained models eq to sequential and cumulative-prioritized replay, it seems misleading to claim the contrary. Precisely, Table 1 compares performance at the last timestep between -- Multi-epoch (90 epochs) is Oracle as it does full passes over the data multiple times, One-Pass (Naive) is sequential and One-Pass (Prioritized Replay) is a variant equivalent to Cumulative-prioritized replay (similar variety to equal/exp). Table 2 extends to Oracles with different compute budgets.
>
> [CB] I am less confident about this, but to the best of my knowledge the equivalence should be as follows -- ERM-Naive is Oracle, Naive is the same as cumulative-all. Minor note: This was also the citation for the issues in expensive sampling strategies point I made.
>
> Overall, both works make the same conclusion made here as far as I see on Imagenet-IID, but have far more extensive analysis (the conclusion is not a new finding). I don't see what does the ImageNet experiment say more than preliminarily confirming the conclusion presented in OPIN (and CB for the increased split case). Both works are more recent than (Ash and Adams, 2020) and specifically have ImageNet-IID experiments same as in this work.

---

> > ### Author Response · Authors · 2023-11-19
> > **Reply to Reviewer fC5h**
> >
> > We thank the reviewer for their detailed reply to our responses. We are very glad to see that we are able to resolve many of the reviewers’ concerns in a satisfactory manner.
> >
> > We are happy to provide additional experiments addressing their remaining concerns. We request reviewer to provide additional clarity on the remaining major concern **[W3.3) Replay buffers not utilized effectively]**.
> >
> > We misinterpreted your comment earlier and confused it with the missing experiment we performed for W2c. We first elaborate and clarify on our method and the motivation for constraining the buffer. Then we elaborate on our understanding of reviewer’s comment to make sure we address it clearly in our followup response.
> >
> > - **Our existing approach of random sampling**:  To restrict the amount of data from previous time steps, we assume that old data is getting progressively expired. In particular, we assume that once we have reduced the data set size from D to D/2, we lose the other D/2 (and can not replay that data in future time). Moreover, when we further reduce the buffer from D/2 to D/3, we subsample the existing dataset with D/2 examples.  Hence we focused on methods that only ablate how we choose which examples to retain when we go from D to D/2, D/2 to D/3 and so on. Here, the motivation is that many companies and regulations require removing expired data from their training set.
> >
> >
> > - We have the following interpretations of the reviewers comment:
> >
> >    **a.** Alter which examples we keep in the buffer (while retaining all old data) as we are training on new time steps so that we can mimic -all with -equal sampling.  In this case, we will not be expiring any data but simply constraining buffer of old data. With equal sampling, we use 1/(t-1) fraction of data for all time steps 1 to (t-1). Thus at time t=3, we use D/2, D/2 and D data (and continue to retain all old data) and at time t=4, we keep D/3, D/3, D/3 and D data (and again continue to retain old data). Here, the D/2 and D/3 examples from old time steps can be completely disjoint. According to this interpretation, we would need to change which 1/(t-1) examples we keep in buffer, to allow more unique examples seen, when we go from say time step t=3 to t=4. Is this understanding correct?
> >
> >    **b.** Do not alter which examples we retain (i.e., keep it the same as our existing approach) but when sampling a specific batch, oversample data from buffers at old time steps to avoid data-imbalance problem. In particular, while sampling every batch with -equal, sample old data with sampling proportion (t-1)/t, so that overall have equal representation of all timesteps in a batch. For example, at time step 3, when we have D/2, D/2 and D data from each time step, we will reweight each of these buffers with proportions 2/3 , 2/3 and 1/3 when creating a new batch.
> >
> >   **c.** Combining the above two corrections.
> >
> > We would appreciate it if the reviewer can clarify if one of our interpretations matches their suggestion regarding W3.3. We clearly understand the other minor comments raised in this round of responses and we are working on addressing those.
> >
> > We thank the reviewer again for their continuous engagement in the discussion.

---

> ### Comment · Reviewer_fC5h · 2023-11-19
> **Clarification**
>
> Hi!
>
> Sorry, maybe that point was too harshly put. The sampling structure that I was thinking about:
>
> **[Memory storage allocation, *not a*)]**: Say in -equal strategy, we would storing D/2 examples from each timestep, and the next iteration D/3 from each past timestep -- here the degree of freedom is how can we choose D/3 out of D/2 samples. My suggestion is prioritize keeping the samples in D/3 which were not/less used for training previously, and break ties by choosing randomly.
>
> To clarify, samples in the D/3 will be subsampled from D/2, not being disjoint. I presume the standard of one loses access to all  samples once discarded, yes!
>
> Now given the buffer, how to choose samples for training-
>
> **[Sample selection from buffer, use b)]** Given that we have fixed the buffer, we know that we have D samples of current timestep and say D/3 samples of each of the past timesteps, we would over-sample D/3 samples compared to current samples, yes. This is avoiding the data imbalance problem. I am relying that *simple augmentations of the past samples like randomflip* while oversampling would mitigate overfitting to a greater extent.
>
> So, not (a) but the sampling I mentioned above which is slightly different from random and (b) I wanted to note augmentation as it might help?
>
> Overall, just wanted to give a strategy which I think mitigates this issue cheaply but I expect to be far better than random sampling, trying to maximize performance. If the authors find clever tweaks (I only spent a few minutes atbest thinking about this at the best), please feel free to make them. I would highlight choosing the oversampling factor was important-- I've found that oversampling by a square root of the proportion worked the best (i.e. if ratio is D/6 old samples in a given timestep to D samples of latest timestep, the sampling is √6:1 between them) but maybe authors find something more principled/better working.

---

> ### Author Response · Authors · 2023-11-22
> **Response to Reviewer fC5h (Part 1)**
>
> We thank the reviewer for their elaborate response and for clarifying the suggestion made for W3.3 about using the count and over-sampling while replaying old data. We ran additional experiments and discuss our results below:
>
>
> > **W1) Sequential and cumulative models behave quite differently …  very happy/convinced by the rebuttal on this point  … would be very happy if I could see the Figure 16 plots across 2-3 timesteps before rebuttal deadline to complete this point.**
>
>
> We thank the reviewer for appreciating our responses on W1. As per your suggestion, we have updated Fig 17 by including heatmaps for TV distances between various reference timesteps, not just 2016. In this revised figure, we can see that as we move away from the reference timestep, the TV distance increases for all reference time steps (represented by different columns). Additionally, we noticed that the TV distance magnitude decreases relatively when later time steps are used as references. We believe that this is due to the inclusion of data from previous years within the 2016 dataset.
>
>
> > **Tasks getting harder over time … Would like to have the bestpool results "For TIC-DataComp (XL), we include results with Bestpool filtering." together in this table or alternatively with and without bestpool filtering in Table 5.**
>
>
> We have updated Table 5 with your suggestion. For the final draft, we will consider integrating this result with Table 2.
>
>
> > **[W3.3) Replay buffers not utilized effectively]. The sampling structure that I was thinking about … [Memory storage allocation, not a)] … [Sample selection from buffer, use b)] … Overall, just wanted to give a strategy which I think mitigates this issue cheaply but I expect to be far better than random sampling, trying to maximize performance.**
>
>
> We have now included results with the suggested method in Table 18 (in Appendix E.2). As the reviewer suggested, we have made the following two modifications: (i) Instead of random sampling, we implemented the count based subsampling that prioritizes not/less used examples; (ii) We oversampled data from old timesteps with ratio inversely proportional to the ratio of examples, i.e., if the old data is of size D/2 and the new data is of size D, then we upsample old data with 2:1 ratio.
>
> However, we observe that this method doesn’t improve performance over Cumulative-equal (and in fact hurts the performance). We hypothesize that this can be due to the decreasing marginal utility of labeled data as highlighted in existing work [1]. Their work argues that due to information overlap among data, as the number of samples increases, the marginal benefit a model can extract from the data diminishes. As a result, Cui et al. [1] proposed using of “effective sample size” instead of actual number of samples to obtain the ratio used to perform re-sampling or re-weighting. We include the expression of effective sample size in more detail in Appendix E.2.
>
> In our settings (even at small scales), our datasets contain an order of 100k img-text pair even after subsampling data from old timestep. For example, with -equal baseline, when training on the last timestep (i.e., 2022), the smallest dataset (i.e., 2016) is of approximately 400k samples.
> Plugging in the expression for effective sample size from Cui et al. [1], we observe that for all $\beta \in (0, 0.99999)$, the ratio of effective sample sizes for different timesteps remains close to 1. This may highlight why our naive over-sampling strategy doesn’t improve over no-oversampling.
>
> Due to a shortage of time and compute, we didn’t perform ablations with $\beta$'s arbitrarily close to 1, but if the reviewer suggests we would be happy to run those experiments for the final version since we have already implemented the oversampling code. We will explore the benefits of count-based subsampling instead of random sampling separately from oversampling in the final version.
>
> ----
>
>
> [1] Cui et al. Class-Balanced Loss Based on Effective Number of Samples. CVPR 2019.

---

> > ### Author Response · Authors · 2023-11-22
> > **Response to Reviewer fC5h (Part 2)**
> >
> > > **All replies except the ImageNet-IID claim seem satisfactory to me … The ImageNet-IID claims seems to misrepresent OPIN and CB, as I explain below …**
> >
> > Thank you for clarifying these referenced papers. We agree with the reviewer that these papers compare with one version of Oracle in their work. We missed the comparison because of confusion in different naming conventions and some minor differences (elaborated below). We have improved the exposition around this in the updated discussion and, for the final draft, we will consider moving the entire paragraph around Imagenet-IID results to the appendix.
> >
> > While we do not claim that our Imagenet-IID results are novel, we believe that our discussion around Imagenet-IID results can be useful to include because of some of the differences highlighted below:
> >
> > *Comparison with OPIN paper*:  We show the performance gap of less than 1% in the same setup used otherwise in the paper when using SOTA training procedures achieving 81% validation performance. Comparitively the referenced OPIN paper does not show whether the 65% to 77% performance gap in Table 1 can be bridged by increasing the compute for their method. Instead, authors show that if they restrict the compute for Oracle in Table 2, the Oracle performance drops to 68% (with $\approx$ 3% gap)
> >
> > *Comparison with CB paper*: As the reviewer pointed out, the most relevant result is in Figure 3 with ERM-naive. However, note that for their comparison, authors perform experiments on DI-Imagenet-2k (Table 2) where they start with an initial memory of Imagenet-1k 1.2 M samples and sequentially observe data for the same classes 1k classes from Imagenet-21k pool. This makes comparing streaming accuracy (or  Imagenet-1k accuracy) for different methods incomparable with our setup (with a gap of over 7% in streaming accuracy even at step 8 as compared to less than 1% in our setup).
> >
> > However, as discussed before, we agree that these papers do discuss Oracle methods in their work and we have now improved this exposition in our paper.
> >
> > We thank the reviewer for their continuous engagement and for sharing detailed thoughts.

---

> ### Comment · Reviewer_fC5h · 2023-11-22
> **Reply to rebuttal**
>
> [W3.3] Interesting, oversampling does not help. Performance degrades due to overfitting. This resolves all my main concerns, I raise my score to 8.
>
> [W Part 2] Yep, resolved! Thanks. I agree that many continual learning setups have slightly different assumptions from each other, preventing direct comparisons.

---

> > ### Author Response · Authors · 2023-11-23
> > **Thank you!**
> >
> > We thank the reviewer for their constructive feedback throughout the discussion period and for helping improve the paper.

---

### Official Review · Reviewer_wUu1 · 2023-11-07

**Soundness:** 2 fair
**Presentation:** 3 good
**Contribution:** 2 fair
**Rating:** 6
**Confidence:** 3

**Summary:**

The paper creates the first set of webscale Time-Continual (TiC) benchmarks for training vision-language models: TIC-DataComp, TIC-YFCC, and TIC-RedCaps with over 12.7B timestamped imagetext pairs spanning 9 years (2014–2022). And they use their benchmarks to curate various dynamic valuations to measure temporal robustness of existing models.

**Strengths:**

The paper collects a large amount of dynamic data to study how to effectively train CLIP models continuously, ensuring the comprehensiveness of the research.

In order to ensure fairness in the evaluation, the paper has established a corresponding experimental protocol.

**Weaknesses:**

The dataset being solely focused on training CLIP may be somewhat limited. Can the article consider incorporating more vision-language models?

The YFCC100M dataset might be somewhat outdated in terms of the years it covers. It may be more representative to explore newer datasets for the research.

**Questions:**

The fundamental issue of continual learning is catastrophic forgetting. If we fine-tune a small number of parameters (e.g., prompt tuning) in the CLIP model, is catastrophic forgetting a major concern? On the other hand, if we fine-tune a large number of parameters, resource limitations may become a factor. Therefore, from this perspective, is it necessary to construct such benchmarks?

---

> ### Author Response · Authors · 2023-11-14
> **Response to Reviewer wUu1**
>
> We thank the reviewer for their feedback and for the positive assessment of our work.
>
> > **The dataset being solely focused on training CLIP may be somewhat limited. Can the article consider incorporating more vision-language models?**
>
> While our paper primarily focuses on experiments with training CLIP models, the constructed benchmark with timestamped image-text pairs can be used to train other Vision Language Models (VLMs). Moreover, since, CLIP vision encoder and text encoder are the basic blocks for other VLMs, e.g., Stable Diffusion models [1] and LLava models [2], we believe that improving CLIP models have direct implications for other VLMs. We would be happy to run additional experiments if the reviewer has any specific suggestions.
>
> > **The YFCC100M dataset might be somewhat outdated in terms of the years it covers. It may be more representative to explore newer datasets for the research.**
>
> We agree with the reviewer that YFCC100M contains data from timestamps before 2014. However, we want to **highlight that we also collected Tic-Datacomp which contains 12.7 B data points till timestamp 2022**, overall giving us data from 2004–2022.
>
>
> > **The fundamental issue … catastrophic forgetting … fine-tune a small number of parameters (e.g., prompt tuning) …  catastrophic forgetting a major concern? …  if we fine-tune a large number of parameters, resource limitations may become a factor … is it necessary to construct such benchmarks?**
>
> In the majority of CL benchmarks, catastrophic forgetting can be attributed to significant and abrupt changes between consecutive timesteps, often across the data distribution, the targets, and the task. For example, the data and targets in perm-MNIST or split-MNIST have little relation to each other from one task to another. Our setup of continual pretraining for foundation models differs from these existing CL benchmarks in two crucial ways: (i)  we observe little catastrophic forgetting; and (ii)  by observing new data we not only benefit on tasks from current time step but also improve performance on tasks from old time steps (these observations are supported with Fig 6 and Fig 4 (right)).
>
> At a high level, by continually training on more data, we hope to improve “general-purpose” abilities of foundation models (e.g., performance on Imagenet). Thus while fine-tuning a limited set of parameters may be an effective strategy for tailoring a model to a confined dataset from a specific domain, this approach is less viable for maintaining and updating general-purpose foundational models  with the dynamically evolving internet data. Given the proportional increase in new data relative to the pre-existing dataset, it necessitates comprehensive model updates.
>
> *(New result)* As per your suggestion, we performed an additional experiment to assess the efficacy of updating selected model parameters versus full-scale fine-tuning with sequential training on the TiC-YFCC dataset. In particular, we compare LiT [1] where we perform full-finetuning on the first time step and then perform LiT for subsequent time steps on Tic-Datacomp (M). We observed that while updating a few parameters does avoid catastrophic forgetting on old timestamps, this happens at the cost of reduced performance on all tasks. For instance, the final Imagenet performance drops substantially from 24.0 (w full finetuning) to 7.3 (w LiT).  This highlights limitations of finetuning only a small number of parameters while trying to keep general purpose foundation models up to date.
>
> We would be happy to answer any further questions you have. If you do not have any further questions, we hope that you might consider raising your score.
>
>
> [1] Zhai et al. 2021. LiT: Zero-Shot Transfer with Locked-image text Tuning

---

> > ### Comment · Reviewer_wUu1 · 2023-11-19
> >
> > Your answer partially solves my questions, but there are still some other weaknesses, such as the lack of sufficient experiments. Therefore, I will keep the score unchanged.

---

> > > ### Author Response · Authors · 2023-11-19
> > > **Reply to Reviewer wUu1**
> > >
> > > We thank the reviewer for their reply. We wanted to get clarity on **"lack of sufficient experiments"**. In the initial review, the following two were the major weaknesses:
> > >
> > > 1. Experiments missing for **"catastrophic forgetting … fine-tune a small number of parameters"**. We performed the LiT experiment to investigate behavior when fine-tuning fewer parameters and observed some limitations on general-purpose abilities. We believe this answers the aforementioned weakness. Let us know if you have any questions about these experiments.
> > >
> > > 2. **"The dataset being solely focused on training CLIP may be somewhat limited. Can the article consider incorporating more vision-language models?".**  We want to clarify that improving CLIP models has direct implications for other VLMs. For example, the Dalle-3 [1] model heavily relies on CLIP for its captioning pipeline.
> > >
> > > Please let us know if you have any specific experimental suggestions for either of these weaknesses. We would be happy to run those. We thank the reviewer again for their continuous engagement.
> > >
> > > ---
> > >
> > > [1] Batker et al. 2023. Improving Image Generation with Better Captions. https://cdn.openai.com/papers/dall-e-3.pdf

---

### Author Response · Authors · 2023-11-14
**Common Response**

We would like to thank the reviewers for their thoughtful feedback. We are glad to see that reviewers appreciated the importance of our benchmark for foundation models (R93w, RTur, fC5h, wUu1), extensive experiments (R93w, fC5h, wUu1), and that the paper is easy-to-follow and well-written (RTur, fC5h).

We are also grateful to the reviewers for their constructive suggestions, which have helped us improve the manuscript. While the initial assessment of the paper includes mixed scores, we are optimistic that we can address key concerns with the updated draft and our responses.

---

### Comment · Area_Chair_pzxh · 2023-11-17
**Author-Reviewer Discussion Phase**

Thank you, reviewers, for your work in evaluating this submission. The reviewer-author discussion phase takes place from Nov 10-22.

If you have any remaining questions or comments regarding the rebuttal or the responses, please express them now. At the very least, please acknowledge that you have read the authors' response to your review.

Thank you, everyone, for contributing to a fruitful, constructive, and respectful review process.

AC

---

### Meta-Review · Area_Chair_pzxh · 2023-12-05

**Metareview:**

This paper introduces the first set of web-scale Time-Continual (TiC) benchmarks for continual learning of vision-language models, specifically CLIP. Time-Continual (TiC) benchmarks consist of multiple datasets with timestamp information, allowing for the evaluation of existing methods on time-continuous data. The proposed benchmarks provide timely and efficient support for researching how to enable large foundational models, particularly CLIP-like vision-language models, to continually learn from new data. Using the proposed benchmarks, the authors conduct an empirical study on efficiently training CLIP models with streaming data. These benchmarks have made significant contributions to the evaluation of large foundational models in continual learning. The analysis delves into existing methods and provides valuable insights for researchers. The overall contribution and significance have been consistently recognized by the reviewers. The authors also provided a very detailed rebuttal, which addressed the raised concerns well. Therefore, I recommend accepting this paper. I also recommend that the authors further revise the paper based on the reviewers' comments.

**Justification For Why Not Higher Score:**

This paper mainly focuses on the rehearsal-based continual learning methods in the experiments, and other various types of continual learning methods are missing, which to some extent limits the applicability of the experimental conclusions in this paper. This shortcoming does affect the further contribution of this paper. I suggest that the authors improve this paper based on the suggestions provided by the reviewers.

**Justification For Why Not Lower Score:**

This paper has conducted enough work to evaluate the continual learning method using a large pretrained vision-language model. The insights and conclusions raised in this paper are valuable. This makes this paper technically sound. Additionally, the authors provide effective rebuttals that address concerns well.

---

### Decision · Program_Chairs · 2024-01-16

Accept (poster)